# Test-time Generalization for Physics through Neural Operator Splitting

Louis Serrano [1 2 3]   Jiequn Han [4]   Edouard Oyallon [5]   Shirley Ho [1 2 4 6]   Rudy Morel [2 4 †]

**The Polymathic AI Collaboration**

## Abstract

Neural operators have shown promise in learning solution maps of partial differential equations (PDEs), but they often struggle to generalize when test inputs lie outside the training distribution, such as novel initial conditions, unseen PDE coefficients or unseen physics. Prior works address this limitation with large-scale multiple physics pretraining followed by fine-tuning, but this still require examples from the new dynamics, falling short of true zero-shot generalization. In this work, we propose a method to enhance generalization at test time, i.e., without modifying pretrained weights. Building on DISCO, which provides a dictionary of neural operators trained across different dynamics, we introduce a neural operator splitting strategy that, at test time, searches over compositions of training operators to approximate unseen dynamics. On challenging out-of-distribution tasks including parameter extrapolation and novel combinations of physics phenomena, our approach achieves state-of-the-art zero-shot generalization results, while being able to recover the underlying PDE parameters. These results underscore test-time computation as a key avenue for building flexible, compositional, and generalizable neural operators. Code is available at https://github.com/LouisSerrano/neural-operator-splitting.

## 1. Introduction

Neural surrogates (de Bézenac et al., 2019; E et al., 2021; Pfaff et al., 2020; Brandstetter et al., 2022) and neural operators (Lu et al., 2021; Li et al., 2020; Kovachki et al., 2021; Raonic et al., 2023; Serrano et al., 2023; Boullé & Townsend, 2024) offer powerful data-driven tools for modeling spatiotemporal dynamics and systems governed by partial differential equations (PDEs). Their main limitation, however, is sensitivity to distribution shifts at test time, i.e., when the dynamics are out-of-distribution (OOD). Such shifts can arise from variations in initial conditions (Chen et al., 2024), error accumulation during autoregressive rollouts (Brandstetter et al., 2022; Lippe et al., 2023; Pedersen et al., 2025), changes in PDE parameters (Kirchmeyer et al., 2022; Koupaï et al., 2024), or fundamentally different underlying dynamics (Takamoto et al., 2022; McCabe et al., 2023; Herde et al., 2024).

We focus on the *parametric setting* (Cohen & Devore, 2015), where a neural surrogate is trained to emulate families of physical dynamics indexed by multi-dimensional coefficient vectors, often corresponding to distinct physical effects. Beyond interpolation within a fixed parameter regime, we are interested in assessing the ability of such surrogates to extrapolate, either to parameter configurations never encountered during training or to novel combinations of physical effects observed only in isolation, while having access to only limited observation data for test-time adaptation.

To address failures in OOD settings, many recent frameworks (McCabe et al., 2023; Herde et al., 2024; Hao et al., 2024) adopt a *pretrain–then–finetune* paradigm. While often effective, this strategy breaks down when very limited data are available for fine-tuning (Koupaï et al., 2024), and the models face fundamental limitations due to their lack of compositionality, with generalization effectively limited to the span of dynamics represented in the pretraining distribution.

Meta-learning (Thrun & Pratt, 1998; Finn et al., 2017) offers an alternative, aiming to learn shared representations that can be rapidly adapted to new parameter regimes (Yin et al., 2022; Kirchmeyer et al., 2022; Koupaï et al., 2024; Nzoyem et al., 2025). However, these approaches have yet to scale reliably to diverse physical systems (Ohana et al., 2024; Morel et al., 2025), and parameter adaptation has been shown to be unstable under distribution shifts (Serrano

---

†Corresponding author. [1]New York University [2]Polymathic AI [3]Emmi AI [4]Flatiron Institute, New York [5]Sorbonne Université, CNRS, ISIR, Paris [6]Princeton University. Correspondence to: Rudy Morel <rmorel@flatironinstitute.org>.

et al., 2025).

To overcome these limitations, we propose a novel test-time adaptation approach based on neural operator splitting. Without modifying the weights of our model, our method approximates test-time dynamics as compositions of operators learned during training, enabling generalization beyond the set of physical phenomena seen during training. The framework consists of three components: (1) a pretrained DISCO model (Morel et al., 2025), a scalable framework that infers a neural operator from each training trajectory and encodes it in a shared, compact latent space; (2) an efficient test-time beam search over the discrete operators discovered during training to identify a suitable decomposition of the unknown dynamics; and (3) operator splitting (Strang, 1968), used both during the search and rollout to approximate the sum of physical terms through successive compositions. Beyond improved test-time adaptation, our method enables system identification by expressing unknown dynamics as compositions of known training operators and is naturally adaptive, requiring less search near the training distribution and more extensive search in far OOD settings.

We evaluate our method against existing approaches on two challenging OOD zero-shot scenarios: when the PDE coefficients lie outside the training distribution, and when the spatiotemporal dynamics result from combinations of physical processes that were observed only individually during training. Our results show that the proposed approach outperforms other methods in both zero-shot settings. Our key contributions are as follows:

- We propose a novel test-time generalization strategy for evolving PDEs based on neural operator splitting to approximate OOD spatiotemporal dynamics.

- We adapt a beam search procedure to combine pretrained operators, balancing accuracy and compute, and provide corresponding test-time scaling laws.

- We demonstrate state-of-the-art zero-shot generalization across diverse nonlinear PDEs on tasks such as parameter extrapolation and operator composition, outperforming adaptive neural operator methods and transformer-based architectures.

- Analysis of the resulting operator decompositions enables system identification and zero-shot PDE parameter estimation.

- To the best of our knowledge, this is the first work to tackle test-time generalization with fixed model weights for predicting PDEs.

## 2. Related Work

**Neural surrogate models and out-of-distribution generalization.** Neural surrogate models have emerged as a powerful tool for accelerating simulation-based workflows for partial differential equations, enabled by the availability of large benchmark datasets and advances in neural operator architectures (Takamoto et al., 2022; Ohana et al., 2024; Koehler et al., 2024; McCabe et al., 2023; Hao et al., 2024; Morel et al., 2025). When trained at scale, these models can achieve high accuracy on in-distribution tasks. However, their performance often degrades sharply when used in out-of-distribution settings, such as unseen PDE parameters, forcing terms, or compositions of physical effects. A response to this challenge is to further scale pretraining by increasing datasets, model and compute size, followed by fine-tuning on OOD target data (Herde et al., 2024; Tripura & Chakraborty, 2026; McCabe et al., 2025). This paradigm implicitly assumes access to enough OOD target samples, which may be unavailable or expensive to obtain in practice.

**Meta-learning for dynamical systems.** To reduce reliance on extensive target data, a second line of work focuses on sample-efficient adaptation through meta-learning. These methods aim to rapidly adapt to new PDEs by exploiting shared structure, but often rely on strong assumptions, such as known PDE coefficients (Wang et al., 2022), symmetry information (Mialon et al., 2023; Sergeant-Perthuis et al., 2022), or restrictive parameterizations (Blanke & Lelarge, 2023). CODA (Kirchmeyer et al., 2022) and GEPS (Koupaï et al., 2024) relax some of these constraints by adapting a shared neural operator to unseen physics, but still requires gradient-based fine-tuning at test time, which can be computationally expensive, especially in far OOD regimes. In contrast, in this paper we design a method that operates at test time without fine-tuning a base model, instead relying on search. Specifically, our method uses the pretrained model to search for an operator that best fits a potentially single target trajectory, without relying on explicit knowledge of the governing equations at test time.

**In-context learning and compositional generalization.** More recently, several works have explored in-context learning and compositional mechanisms for differential equations. ICON (Yang et al., 2023) demonstrates in-context prediction for ODEs by conditioning on example trajectories of related systems, enabling adaptation without parameter updates. Cao et al. (2024); Serrano et al. (2025); Koupaï et al. (2025) extend this idea to PDEs, showing that models can in some cases generalize in context by conditioning on solutions from related tasks. However, the mechanisms by which they implicitly compose or reuse physical operators often remain implicit and difficult to access. For example, recent work by Fear et al. (2025) rely on identifying hid-

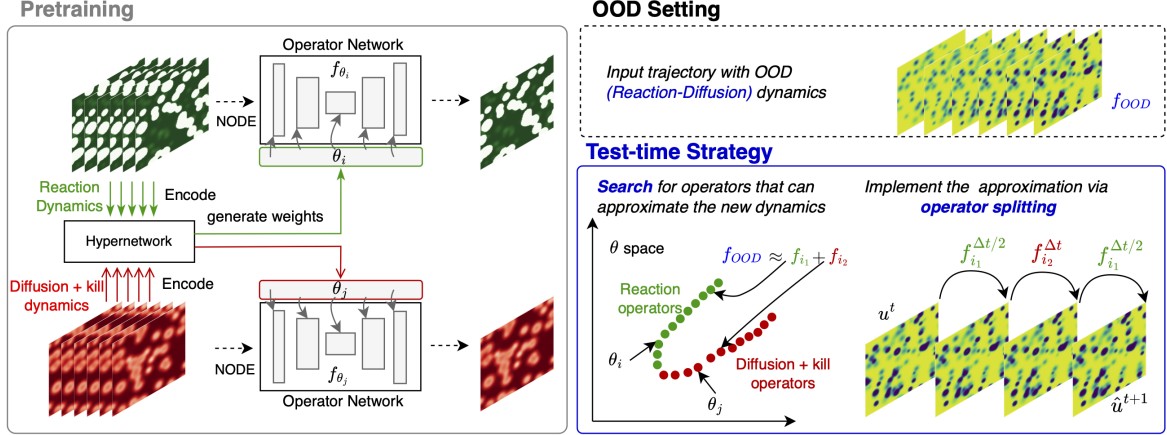

*Figure 1.* **Test-time generalization through neural operator splitting**. During pretraining (left), DISCO learns a dictionary of operators for different physics, such as reaction dynamics (green) and diffusion+kill dynamics (red), with a hypernetwork generating corresponding operator weights $\theta_i, \theta_j$. At test time (right), on OOD dynamics such as reaction-diffusion, our method searches over combinations of these operators to approximate the new dynamics (e.g., $f_{OOD} \approx f_{i_1} + f_{i_2}$), and evolves $u^t \to u^{t+1}$ using neural operator splitting via sequential operator applications.

den steering vectors tied to specific physical phenomena, requiring careful post hoc probing of model activations. In our paper, we instead rely on the explicit composition of neural operators in order to provide an adaptation strategy to OOD settings without requiring training on target data. Our method searches over compositions of pretrained operators identified during training by DISCO (Morel et al., 2025) and selects the one that best fits an observation of the OOD target trajectory. To our knowledge, this is the first work to bring search-based adaptation and test-time compute scaling to neural PDE solvers.

## 3. Problem Setting

Data-driven models for evolving unknown PDEs are typically trained on trajectories with varying PDE class, coefficients, and initial conditions, with the goal of generalizing to unseen scenarios. While prior work has generally emphasized generalization over novel initial conditions under a fixed PDE, we consider the additional challenge of generalizing to unseen PDE coefficients or even entirely new PDE classes. This setting relates to approaches such as MPP (McCabe et al., 2023), DISCO (Morel et al., 2025) or WALRUS (McCabe et al., 2025), but we restrict the diversity of training physics to better evaluate OOD generalization.

**Parametric PDE setting.** We consider a family of parametric PDEs of the form

$$\partial_t u = \sum_{k=1}^{K} \mu_k \, \mathcal{F}_k(u, \nabla_x u, \nabla_x^2 u, \dots),$$

where $u(x,t)$ is the solution field, $\mu = (\mu_1, \dots, \mu_K) \in \mathcal{M}$ is a parameter vector, and $\{\mathcal{F}_k\}_k$ denote fundamental physics operators (e.g., advection, diffusion, reaction, etc.). During training, parameters are drawn from a *sparse distribution* $P^{\text{train}}(\mu)$, where only one operator is present for each trajectory. Concretely, each sample takes the form $\mu = (0, \dots, \mu_k, \dots, 0)$, with exactly one nonzero component $\mu_k \in \mathcal{M}_k^{\text{train}}$, restricted to a prescribed training range.

**OOD challenges.** This training setup naturally induces two distinct types of OOD scenarios at test time:

- *Parameter Extrapolation*: Parameters remain sparse but take values *outside* the convex hull of training ranges: $\mu_{\text{test}} = (0, \dots, \mu_k^{\text{test}}, \dots, 0)$ with $\mu_k^{\text{test}} \notin \text{conv}(\mathcal{M}_k^{\text{train}})$.

- *Operator Composition*: Multiple operators are simultaneously present, though each parameter still lies *within* its training range: $\mu_{\text{test}} = (\mu_1, \dots, \mu_K)$ with several $\mu_k \neq 0$ and $\mu_k \in \text{conv}(\mathcal{M}_k^{\text{train}})$.

For illustration, consider the advection–diffusion equation $\partial_t u + c\,\partial_x u = D\,\partial_{xx} u$, with advection speed $c$ and diffusion coefficient $D$. Training covers pure advection ($c \in [0,1], D = 0$) and pure diffusion ($c = 0, D \in [0,1]$) separately. At test time, parameter extrapolation may involve $c = 2.5, D = 0$, while operator composition may involve $c = 0.5, D = 0.3$.

**Zero-shot prediction task.** Given this OOD setting, our task is to predict rollout trajectories in a zero-shot manner using only the observed dynamics at test time. Specifically,

we observe $L$ consecutive snapshots of a test trajectory $u_{\text{test}}^{1:L}$ with temporal discretization $\Delta t$, which characterize the underlying dynamics that were never seen during training. From these observations alone, we must best predict the subsequent $H$ snapshots $\hat{u}_{\text{test}}^{L+1:L+H}$ without training on this specific system. Performance is evaluated using the normalized relative mean squared error (NRMSE) against the ground truth, computed over space and time:

$$\text{NRMSE}(u_{\text{test}}, \hat{u}_{\text{test}}) = \frac{||u_{\text{test}} - \hat{u}_{\text{test}}||_2}{||u_{\text{test}}||_2} \, .$$

# 4. Method

As illustrated in Figure 1, our method leverages a DISCO-pretrained model that learns a dictionary of neural operators from the training data. When presented with a single out-of-distribution trajectory at test time, we search for combinations of these operators that best explain the new dynamics, while keeping all model parameters fixed. This combination is realized through *neural operator splitting*.

## 4.1. Constructing a Dictionary of Operators

The DISCO framework (Morel et al., 2025) learns to predict PDE evolution by discovering appropriate differential operators from trajectory context. It consists of two main components: a hypernetwork $\psi_\alpha$ that processes spatiotemporal context, and a small operator network $f_\theta$ that performs the actual time integration. Given a trajectory context $u^{1:L}$, DISCO operates through:

$$\hat{u}^{L+1} = u^L + \int_L^{L+1} f_\theta(u^t)\, dt, \quad \text{with} \quad \theta = \psi_\alpha(u^{1:L}),$$

where $\psi_\alpha$ is a transformer with learnable parameters $\alpha$, and $f_\theta$ is a U-Net whose parameters $\theta$ are dynamically generated by the hypernetwork. After pretraining, we extract a dictionary of neural operators by encoding each trajectory $i$ from the training set: $\{f_{\theta_i} = \psi_\alpha(u_i^{1:L})\}$. To simplify notation, we denote $f_i = f_{\theta_i}$. This dictionary of operators $f_1, \ldots, f_N$ will form the foundation of our test-time search strategy.

While we instantiate the dictionary using DISCO, the search and underlying splitting procedures presented next are framework- and architecture-agnostic, applying to any backbone that produces a dictionary of trajectory-conditioned neural-ODE operators.

## 4.2. Operator Composition Search

Given a test trajectory $u_{\text{test}}^{1:L}$ governed by unknown, potentially OOD dynamics, our goal is to approximate the underlying system by composing operators from our dictionary $\{f_1, \ldots, f_N\}$. We seek a subset $S = \{f_{i_1}, f_{i_2}, \ldots, f_{i_m}\}$

such that the sum $f_{i_1} + f_{i_2} + \cdots + f_{i_m}$ best approximates the test dynamics. In practice, this sum is implemented through operator splitting as detailed in Section 4.3.

**Optimization objective.** We define $\mathcal{L}(S) = \frac{1}{L-1}\sum_{t=1}^{L-1} \text{NRMSE}(u_{\text{test}}^{t+1}, \hat{u}_{\text{test}}^{t+1})$ as the fitting error when using the operator subset $S$, where $\hat{u}_{\text{test}}^{t+1}$ is the prediction obtained by applying operator splitting with the operators in $S$ starting from $u_{\text{test}}^t$. Our test-time adaptation seeks the subset that minimizes this objective $S^* = \arg\min_{S \subseteq \{f_1, \ldots, f_N\}} \mathcal{L}(S)$.

**Search strategies.** Since an exhaustive search over all $2^N$ possible subsets is intractable, we investigate two complementary strategies that balance exploration with computational efficiency.

*Uniform Sampling*: As a baseline, we uniformly sample subsets of size $m \sim \text{Uniform}(1, M)$ by drawing $m$ operators from our dictionary, where $M$ is a small maximum subset size. We evaluate $T$ trials, giving a computational complexity of $O(T)$ and then keep the subset with the lowest objective error. See Algorithm 2 for more details.

*Beam Search*: We use beam search to greedily explore operator compositions while maintaining exploration. Starting with the top-$B$ single operators ($B$ is the beam width), we iteratively expand each candidate by adding one more operator and keep only the $B$ best combinations:

$$\mathcal{B}_0 = \text{top-}B \text{ operators from } \{f_1, \ldots, f_N\},$$
$$\mathcal{B}_{m+1} = \text{top-}B \text{ from } \{S \cup \{f_j\} : S \in \mathcal{B}_m\}.$$

Here, $\mathcal{B}_0$ contains singletons, $\mathcal{B}_1$ pairs, $\mathcal{B}_2$ triples, and so on. The computational complexity is $O(BN)$ candidate evaluations per iteration, with $N$ the size of the dictionary. When $B = 1$, this reduces to greedy sequential selection. To prevent excessive operator compositions, we impose both a minimum relative improvement threshold to continue the search and a maximum composition length of $M$. The pseudo-code is detailed in Algorithm 1.

## 4.3. Operator Splitting for Neural Operators

To implement the sum $f_{i_1} + f_{i_2} + \cdots + f_{i_m}$ in practice, we employ operator splitting of the neural operators, a technique that we coin *neural operator splitting*. For two operators $f_1 + f_2$, Lie splitting sequentially applies each operator over the full time step: $\hat{u}^{L+1} = f_2^{\Delta t} \circ f_1^{\Delta t}(u^L)$, where $f_i^{\Delta t}$ represents integrating operator $f_i$ for time $\Delta t$. Strang splitting uses a symmetric pattern for higher accuracy: $\hat{u}^{L+1} = f_1^{\Delta t/2} \circ f_2^{\Delta t} \circ f_1^{\Delta t/2}(u^L)$. This reduces the approximation error from $\mathcal{O}(\Delta t^2)$ to $\mathcal{O}(\Delta t^3)$ (Strang, 1968; Holden et al., 2010). For multiple operators, we extend these patterns and refer to Section B.3 for additional

*Table 1.* **Zero-shot generalization to unseen PDE combinations.** Average NRMSE over $H$ predicted steps (lower is better) on unseen combinations of physical phenomena. During pretraining, models see each phenomenon individually (e.g., pure diffusion or Euler). At test time, multiple phenomena appear simultaneously (e.g., Navier–Stokes combines Euler with diffusion). With fixed weights (no fine-tuning), standard models struggle to generalize, whereas our adaptive operator splitting achieves substantial gains without retraining.

| Method | Advection + Diffusion | Nonlinear Adv. + Diffusion | Nonlinear Adv. + Dispersion | Diffusion + Dispersion | All Three | Reaction + Diffusion | Euler + Diffusion |
|---|---|---|---|---|---|---|---|
| MPP | 0.270 | 0.050 | 0.105 | 0.091 | 0.128 | 0.191 | 0.273 |
| FNO | 0.318 | 0.031 | 0.165 | 0.038 | 0.129 | 0.224 | 0.241 |
| Zebra | 0.893 | **0.022** | 0.241 | 0.069 | 0.193 | 0.127 | 0.198 |
| GEPS | 0.039 | 0.039 | 0.249 | 0.229 | 0.265 | 0.128 | 0.786 |
| DISCO (Original) | 0.170 | 0.085 | 0.100 | 0.120 | 0.164 | 0.245 | 0.572 |
| Ours (Uniform) | 0.043 | 0.068 | 0.103 | 0.043 | 0.075 | 0.089 | 0.209 |
| Ours (Beam) | **0.015** | 0.056 | **0.049** | **0.007** | **0.036** | **0.089** | **0.066** |

details. This is to the best of our knowledge the first application of operator splitting in the context of neural PDE surrogates.

## 5. Experiments

We evaluate our test-time search strategy on two challenging OOD scenarios, using distinct benchmarks to systematically assess the capabilities of operator composition and test-time adaptation.

We begin by describing the experimental setup and training dataset (Section 5.1). We then evaluate extrapolation performance to unseen PDE parameter ranges, demonstrating how test-time search enables robust generalization beyond the training distribution (Section 5.2). Next, we assess our method's ability to handle novel compositions of physical processes in Section 5.3. Finally, in Section 5.4, we analyze how our approach benefits from increased computational budget during test-time search, showing consistent performance improvements, and demonstrate its capacity for parameter identification in previously unseen dynamics.

### 5.1. Experimental Setting

**Datasets.** We design four benchmark datasets that systematically evaluate compositional generalization capabilities across different physics regimes and spatial dimensions. Each training dataset enforces strict separation of physical processes as described in Section 3, with operators learned exclusively from trajectories containing individual physics components, never their combinations.

**1D Advection-Diffusion.** Our first benchmark focuses on linear transport phenomena governed by $\frac{\partial u}{\partial t} = D\frac{\partial^2 u}{\partial x^2} - c\frac{\partial u}{\partial x}$ on a periodic domain of length $l = 16$ with 256 spatial discretization points. Training data consist exclusively of single-physics trajectories: pure advection with speeds $c \in [0.01, 1.0]$ and zero diffusion ($D = 0$), or pure diffu-

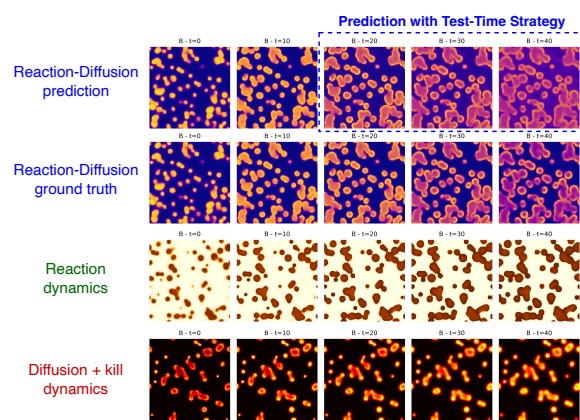

*Figure 2.* **Test-time generalization on Gray–Scott equations.** Our neural operator search correctly predicts an unseen, non-trivial dynamics (compare first and second rows), which differs substantially from the reaction dynamics (third row) or diffusion+kill dynamics (fourth row) seen during training, demonstrating that our method, based on combining simple operators, can capture complex phenomena. The dashed blue box represents predictions with our strategy, and snapshots at t=0 and t=10 are representative of the $L = 16$ context frames used as input for adaptation.

sion with coefficients $D \in [0.001, 1.0]$ and zero advection ($c = 0$). Each trajectory contains 100 temporal snapshots spanning $T = 10$ seconds.

**1D Combined Equation.** The second dataset examines the nonlinear advection-diffusion-dispersion equation: $\frac{\partial u}{\partial t} + \alpha\frac{\partial u^2}{\partial x} - \beta\frac{\partial^2 u}{\partial x^2} + \gamma\frac{\partial^3 u}{\partial x^3} = 0$, where $\alpha$, $\beta$, and $\gamma$ quantify the strength of nonlinear advection, diffusion, and dispersion effects, respectively. Training isolates each physical mechanism with parameter combinations $(\alpha, 0, 0)$, $(0, \beta, 0)$, and $(0, 0, \gamma)$, where coefficients are sampled uniformly from $\alpha \in [0, 1]$, $\beta \in [0, 0.4]$, and $\gamma \in [0, 1]$. We generate 8,192 training trajectories for each physics across 128 parameter configurations, with each trajectory containing 250 temporal

snapshots on a 256-point spatial grid over $T = 4$ seconds and a periodic domain of length $l = 16$. We employ the solver from (Brandstetter et al., 2022) to generate the trajectories.

**2D Reaction-Diffusion.** The third benchmark is the Gray–Scott reaction-diffusion equation from The Well (Ohana et al., 2024):

$$\frac{\partial A}{\partial t} = D_A \nabla^2 A - \delta A B^2 + F(1 - A) \,,$$
$$\frac{\partial B}{\partial t} = D_B \nabla^2 B + \delta A B^2 - (F + k)B \,.$$

This system models the spatiotemporal evolution of two chemical species parameterized by diffusion coefficients $D_A, D_B$, reaction strength $\delta$, feed rate $F$ for species $A$, and kill rate $k$ for species $B$. We construct training data using two operator types: (1) diffusion-kill operators with fixed diffusion coefficients $D_A = 2 \times 10^{-5}$, $D_B = 1 \times 10^{-5}$, disabled reaction ($\delta = 0$, $F = 0$), and kill rates $k$ spanning 20 values in $\{0.051, 0.052, 0.053, \ldots, 0.069, 0.070\}$; (2) pure reaction operators with disabled diffusion ($D_A = D_B = 0$), unit reaction strength ($\delta = 1$), zero kill rate ($k = 0$), and feed rates $F$ taking 20 values in $\{5, 10, \ldots, 95, 100\} \times 10^{-3}$. The spatial domain employs a $128 \times 128$ grid with periodic boundary conditions. We generate 512 trajectories per parameter configuration, using clustered gaussians as initial conditions, simulating for $T = 50$ and keeping 50 snapshots.

**2D Navier–Stokes.** Our most challenging benchmark considers a two-dimensional fluid dynamics benchmark based on the evolution of the vorticity field $\omega(t, x, y)$ on a periodic square domain $[0, 2\pi]^2$, governed by the incompressible Navier–Stokes equations in vorticity form:

$$\partial_t \omega + \mathbf{u} \cdot \nabla \omega = \nu \Delta \omega,$$
$$\mathbf{u} = (-\partial_y \psi, \partial_x \psi), \quad -\Delta \psi = \omega.$$

This formulation combines conservative advection and viscous diffusion. We construct training data using two operator types: (1) *pure advection* operators corresponding to the inviscid Euler equations ($\nu = 0$), and (2) *pure diffusion* operators governed by the heat equation $\partial_t \omega = \nu \Delta \omega$.

Trajectories are simulated on a $512 \times 512$ grid over a time horizon $T = 4.0$, and we keep 50 snapshots, spectrally downsampled to $256 \times 256$ for learning. Diffusion viscosities are sampled at 16 logarithmically spaced values in $[10^{-4}, 10^{-2}]$, with the dataset balanced across viscosities.

**Implementation.** We use the following hyperparameters for our test-time search strategies. **Uniform:** $T = 100$ combinations for advection-diffusion, combined equation, and Navier–Stokes, $T = 200$ for Gray–Scott, with maximum composition length $M = 4$. **Beam:** We subsample $N$

operators per benchmark: 256 for advection–diffusion, 96 for the combined equation, 40 for Gray–Scott, and 17 for Navier–Stokes. We use beam width $B = 4$ for advection-diffusion, combined equation, Navier–Stokes, $B = 8$ for Gray–Scott, maximum composition length $M = 5$, and improvement threshold of 5%. We study sensitivity to the dictionary size $N$ and the fitting-window length $L$ in Appendix C.5 and Appendix C.6, finding the choices used here to be robust.

**Baselines.** We compare against different state-of-the-art approaches. All methods are trained from scratch on the same training datasets designed for this study. We use next-step prediction as the learning objective. **DISCO (Original)** (Morel et al., 2025): We validate that our framework systematically improves upon the original DISCO approach, which performs predictions by directly encoding out-of-distribution trajectories and predicting dynamics without test-time adaptation. **FNO** (Li et al., 2021): We include the classic Fourier Neural Operator as a representative non-adaptive neural operator that learns the solution map in spectral space. **MPP** (McCabe et al., 2023): We compare against the Axial Vision Transformer (Ho et al., 2019) architecture designed for multiple physics pretraining, representing state-of-the-art performance in large-scale physics foundation models. **Zebra** (Serrano et al., 2025): We include this autoregressive transformer inspired by language modeling. While primarily designed for one-shot and few-shot adaptation for in-context learning, Zebra provides a valuable comparison as a generative model that requires higher computational resources than deterministic models. **GEPS** (Koupaï et al., 2024): We also compare against this meta-learning framework designed for efficient few-shot adaptation to changing dynamics. GEPS is trained using an environment-based perspective (Yin et al., 2022; Kirchmeyer et al., 2022), and employs a LoRA-based adaptation scheme (Hu et al., 2021), making it a suitable comparison point for parameter efficient fine-tuning approaches.

**Test evaluation.** We evaluate all methods by unrolling predictions for $H$ steps per benchmark: 34 for advection–diffusion, 50 for the combined equation, 32 for Gray–Scott, and 16 for Navier–Stokes. All experiments use a history of $L = 16$ snapshots as context, either for direct prediction (FNO, MPP, Zebra, original DISCO) or for adaptation (GEPS and our framework). We report the average NRMSE over the entire predicted trajectory as the primary evaluation metric.

### 5.2. Parameter Extrapolation

**Setting.** In this section, we investigate extrapolation capabilities on advection-diffusion systems by testing higher advection speeds $c \in [1, 3]$ and higher diffusion coefficients

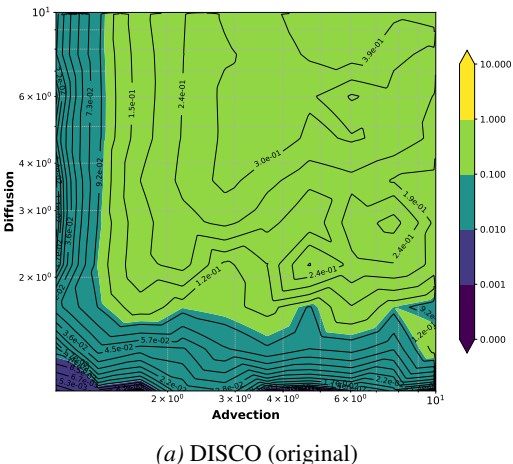

*(a)* DISCO (original)

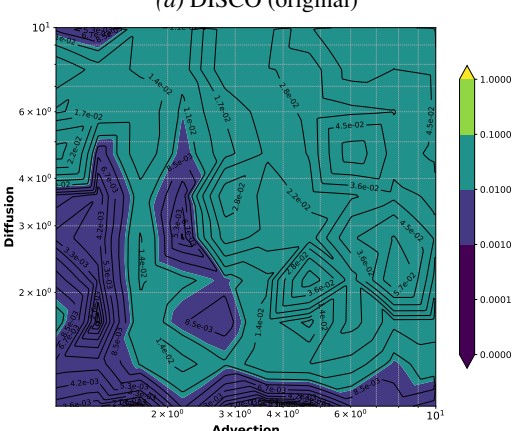

*(b)* Beam search

*Figure 3.* **Zero-shot generalization to unseen advection-diffusion PDEs.** Contour maps report the NRMSE (averaged over 34 rollout steps; lower is better) over the grid of advection (x-axis) and diffusion coefficients (y-axis). (a) DISCO predictions obtained with the operator produced by the encoder and hypernetwork. (b) Our test-time method: beam search selects operators from the dictionary to minimize the fitting error; the selected operators are then composed and unrolled via neural operator splitting.

$D \in [1, 3]$. While higher advection speeds present significant challenges for classical numerical solvers due to transport dominance, higher diffusion coefficients generally provide better numerical stability through smoothing effects. We also examine extrapolation performance for the nonlinear advection term $\alpha \in [1, 2]$ and dispersion coefficient $\gamma \in [1, 2]$ in the combined equation.

**Results.** Table 2 shows that our test-time operator composition strategy consistently improves upon the original DISCO by orders of magnitude across all benchmarks, demonstrating the generalization capabilities of neural operator splitting compared to direct out-of-distribution encoding. The beam search variant achieves the strongest performance, with improvements ranging from 10× on advection speed extrapolation to over 200× on diffusion coefficient

*Table 2.* **Zero-shot performance on PDE parameter extrapolation.** Average NRMSE over $H$ predicted steps (lower is better).

| | Adv.–Diff. | | Combined | |
| --- | --- | --- | --- | --- |
| **Method** | $c$ | $D$ | $\alpha$ | $\gamma$ |
| MPP | 0.588 | 0.409 | 0.134 | 0.369 |
| FNO | 0.492 | 0.166 | 0.166 | 0.317 |
| Zebra | 1.070 | 1.579 | 0.128 | 0.448 |
| GEPS | 0.848 | 0.267 | 0.020 | 0.782 |
| DISCO | 0.768 | 0.159 | 0.088 | 1.007 |
| Ours (Uniform) | 0.113 | 0.055 | 0.027 | 0.070 |
| Ours (Beam) | **0.052** | **0.002** | **0.016** | **0.022** |

tasks. Among the baselines, GEPS shows competitive performance on nonlinear advection and reasonable results on diffusion tasks, but exhibits instability on dispersion extrapolation. We observed that GEPS fine-tuning often leads to diverging operators during rollout for these OOD settings, consistent with previous findings on gradient-based adaptation methods (Serrano et al., 2025). Zebra particularly struggles on advection-diffusion tasks, which we attribute to limitations in discrete tokenization for capturing the high-frequency dynamics present in our fractal-based initial conditions. MPP and FNO provide robust but modest performance across tasks, never excelling in these extrapolation scenarios.

### 5.3. Physics Composition

**Setting.** We then evaluate the compositional capabilities of neural operators by testing their ability to combine previously isolated physical processes. For the advection-diffusion system, the test cases combine both mechanisms with coefficients sampled uniformly from $c \in [0, 1]$ and $D \in [0, 1]$.

For the combined equation dataset, we test four types of multi-physics compositions: (1) nonlinear advection + diffusion with $\alpha \in [0, 1], \beta \in [0, 0.4]$; (2) nonlinear advection + dispersion with $\alpha \in [0, 1], \gamma \in [0, 1]$; (3) diffusion + dispersion with $\beta \in [0, 0.4], \gamma \in [0, 1]$; and (4) all three processes combined with $\alpha \in [0, 1], \beta \in [0, 0.4], \gamma \in [0, 1]$.

For the Gray–Scott system, we assess compositional generalization using test trajectories spanning the full parameter space induced by the Cartesian product of feed rates ($F$) and kill rates ($k$) from the training distribution, recombining reaction and diffusion dynamics that were seen separately.

Finally, we evaluate whether models pretrained on Euler and Diffusion can generalize to their composition in the 2D Navier–Stokes equations, using matched viscosity values.

**Results.** Table 1 shows that our method achieves the best performance on 6 out of 7 unseen composition tasks, with consistent improvements over the original DISCO approach.

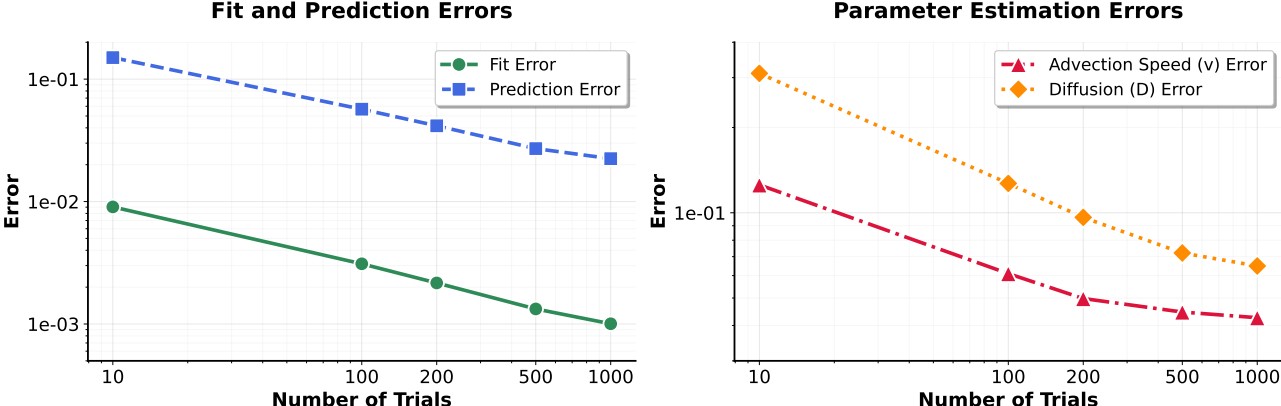

*Figure 4.* **Test-time scaling laws.** (**Left**) Fitting and prediction error (mean NRMSE over 34 rollout steps) of our test-time method with uniform search on an out-of-distribution (OOD) advection–diffusion dynamics, as a function of the number of uniform-search trials. (**Right**) PDE parameter identification error (mean absolute error, MAE) versus the number of trials, showing that the selected operators recover meaningful and accurate physical parameters.

These gains are most pronounced for more complex compositions involving nonlinear or higher-order interacting terms, where directly encoding out-of-distribution trajectories often produces inaccurate or unstable operators. The beam search variant consistently outperforms uniform sampling, indicating that efficiently identifying a small set of compatible operators is more effective than evaluating many unstructured combinations.

Among baselines, we observe clear structural trends. MPP provides robust performance across tasks, likely due to the stability of its vision-based encoder, but lacks the compositional flexibility required to accurately model coupled dynamics. Zebra performs well on most compositions, particularly those involving nonlinear advection, reflecting the flexibility of its autoregressive formulation, though its performance degrades on more complex regimes. GEPS achieves competitive results on simpler compositions but becomes unstable as compositional complexity increases, consistent with the sensitivity of gradient-based test-time adaptation under distribution shift.

We further assess the framework's robustness in Appendix C.2 (test dynamics containing perturbation terms not representable by any dictionary operator) and Appendix C.3 (training trajectories that already mix multiple physical mechanisms), and isolate the contribution of the DISCO training recipe via an ablation in Appendix C.4.

Figures 2–16 complement the numerical results with qualitative rollouts, showing that our method qualitatively captures coupled dynamics in Gray–Scott and Navier–Stokes. Figure 3 further demonstrates consistent improvements of the beam search over DISCO for advection–diffusion across the full coefficient range.

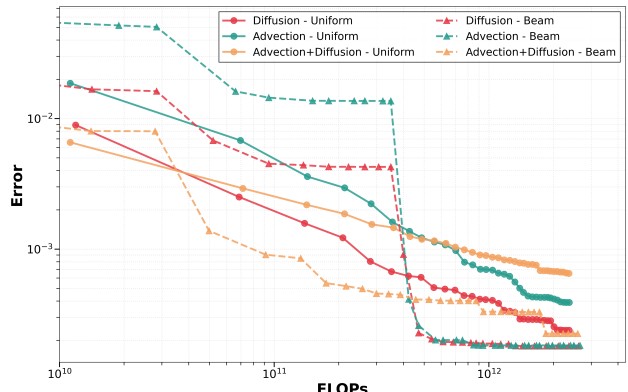

*Figure 5.* **Fitting error versus computational cost (FLOPs)** for uniform and beam search strategies across three tasks: diffusion extrapolation, advection extrapolation, and combined advection–diffusion. Beam search proceeds sequentially over operator complexity, first selecting the best single operators, and subsequently higher-order compositions. At intermediate FLOP budgets, beam search achieves substantially lower error than uniform search, while uniform search exhibits a smooth, approximately power-law decay with increasing computation.

### 5.4. Compute Scaling Analysis

**Test-time scaling laws.** Figure 4 shows that increasing the number of uniform-search trials consistently reduces both fitting and prediction error, exhibiting a power-law-like decay. Moreover, improved fitting error strongly correlates with better rollout accuracy. To highlight the computational advantage of beam search over uniform search, Figure 5 plots fitting error as a function of cumulative compute (total FLOPs). Beam search rapidly outperforms uniform search by progressively expanding the operator set—first searching over single operators, then pairs, then triples, and so on—which produces sharp drops in error as more expres-

sive combinations become available. In contrast, uniform search improves smoothly but more slowly, indicating lower compute efficiency. A wall-clock comparison against GEPS gradient-based fine-tuning, on a single A100, is reported in Appendix C.7: beam search is competitive with extensive GEPS adaptation on advection–diffusion and remains stable on far-OOD extrapolation, where GEPS diverges.

**Parameter identification.** Beyond predictive accuracy, our method supports interpretable test-time parameter identification by tracing each selected operator to the training trajectory from which it was produced, and using that trajectory's known physical coefficients. We estimate the coefficients of a new test trajectory by summing the parameter contributions associated with the selected operators. As shown in the right panel of Figure 4, estimation accuracy improves as the search progresses for both the advection speed and diffusion coefficient in the advection–diffusion setting. Notably, despite never being trained on the coupled system, our method is able to recover accurate parameters for PDEs that combine these distinct physical effects. This parameter-recovery behavior parallels classical data-driven equation-discovery methods such as SINDy (Brunton et al., 2016), though parameters are recovered through compositional search over learned operators rather than through sparse regression over a symbolic library of handcrafted components.

## 6. Conclusion

We introduced a method for generalizing to out-of-distribution dynamical systems at test time without modifying model weights. Given a dictionary of neural operators identified at training time by a pretrained DISCO model (Morel et al., 2025), our method searches for combinations of these operators that best explain out-of-distribution trajectories, enabling zero-shot generalization. Complementary to fine-tuning strategies, our method offers an additional direction for test-time generalization relying on neural operator composition and neural operator splitting. We showed empirically that this approach achieves state-of-the-art performance on zero-shot out-of-distribution dynamics including the Navier–Stokes equations.

Current limitations include the requirement that composed operators share compatible input and output domains, and the wall-clock overhead of test-time search (scaling as $\mathcal{O}(B \cdot N \cdot L)$ operator forward passes per beam-search iteration; see Appendix C.7). The search overhead is competitive with gradient-based fine-tuning in our 1D and 2D experiments but may become a bottleneck in 3D, where individual forward passes are more expensive. Extending neural operator splitting to broader forms of compositionality across spatial dimensionality, discretizations, and physical

fields, and scaling the search efficiently to 3D, represents a promising direction for future work.

## Acknowledgements

The authors thank the Scientific Computing Core at the Flatiron Institute, a division of the Simons Foundation, for providing computational resources and support. They also thank Francesco Pio Rammuno, Alex Nguyen, Tanya Marwah, and Patrick Gallinari for insightful discussions.

We would like to acknowledge the support of the Simons Foundation and of Schmidt Sciences. This work was supported in part by the AI2050 program at Schmidt Sciences (Grant G-25-70028)

EO acknowledges funding from PEPR IA (grant SHARP ANR-23-PEIA-0008). He was granted access to the AI resources of IDRIS under the allocation 2025-AD011015884R1.

## Impact Statement

This paper presents work whose goal is to advance the field of Machine Learning. There are many potential societal consequences of our work, none which we feel must be specifically highlighted here.

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

# A. Dataset Details

## A.1. Advection-diffusion

We generate synthetic trajectories for the 1D advection-diffusion equation

$$\frac{\partial u}{\partial t} + v\frac{\partial u}{\partial x} = D\frac{\partial^2 u}{\partial x^2} \tag{1}$$

with periodic boundary conditions, where $v$ is the advection speed and $D$ is the diffusion coefficient. The dataset uses analytical solutions computed via Fourier spectral methods to avoid numerical errors.

**Physical Parameters.** During training, we generate 50% pure advection cases ($v \sim \text{Uniform}(0.01, 1.0)$, $D = 0$) and 50% pure diffusion cases ($v = 0$, $D \sim \text{Uniform}(0.001, 1.0)$).

**Initial Conditions.** We generate complex initial conditions using Fractaloid with random phase patterns, which create self-similar signals with power-law spectra. These patterns are constructed as trigonometric polynomials

$$u_0(x) = \sum_{k=1}^{\text{degree}} a_k k^{-\text{power}} \sin(k\theta + \phi_k), \tag{2}$$

where $a_k$ are independent Gaussian coefficients and $\phi_k$ are random phases. We use degree $= 256$ and power is sampled uniformly in $[1, 4]$, then normalize each initial condition to zero mean and unit variance. For testing, we fix the power to 3.

**Analytical Solutions.** We compute exact solutions using Fourier spectral methods. In spectral space, the solution evolves as $\hat{u}(k, t) = \hat{u}_0(k)\exp(-Dk^2 t)\exp(-ikvt)$, which we transform back to physical space via inverse FFT. The spatial domain has length $L = 16.0$ with $n_x = 256$ grid points, evolved over $n_t = 100$ time steps to final time $T = 10.0$.

## A.2. Combined-equation

We follow the dataset generation approach of (Brandstetter et al., 2022), with key distinctions in the physics formulation for training data generation and the exclusion of forcing terms. The combined equation is governed by the following PDE:

$$\partial_t u + \partial_x(\alpha u^2 - \beta\partial_x u + \gamma\partial_{xx}u) = 0, \tag{3}$$

subject to periodic boundary conditions and initial conditions

$$u_0(x) = \sum_{j=1}^{J} A_j \sin(2\pi\ell_j x/l + \phi_j). \tag{4}$$

This formulation combines three fundamental physical mechanisms: nonlinear advection ($\alpha u^2$), linear diffusion ($-\beta\partial_x u$), and dispersion ($\gamma\partial_{xx}u$). For each initial condition, we sample the Fourier mode coefficients: $A_j \sim \text{Uniform}([-0.5, 0.5])$, $\ell_j \sim \text{Uniform}(\{1, 2, 3, 4, 5\})$, and $\phi_j \sim \text{Uniform}([0, 2\pi])$ with $J = 5$ modes.

**Training dataset** The training data are generated using parameter combinations $(\alpha, 0, 0)$, $(0, \beta, 0)$, and $(0, 0, \gamma)$. The coefficients are sampled uniformly from $\alpha \in [0, 1]$, $\beta \in [0, 0.4]$, and $\gamma \in [0, 1]$. We generate 8,192 training trajectories for each isolated physics across 128 parameter configurations. Each trajectory contains 250 temporal snapshots on a 256-point spatial grid over $T = 4$ seconds with a periodic domain of length $l = 16$.

## A.3. Reaction-Diffusion

Our most challenging benchmark is the Gray–Scott reaction-diffusion system from The Well (Ohana et al., 2024):

$$\frac{\partial A}{\partial t} = D_A\nabla^2 A - \delta AB^2 + F(1 - A), \tag{5}$$

$$\frac{\partial B}{\partial t} = D_B\nabla^2 B + \delta AB^2 - (F + k)B. \tag{6}$$

This system models the spatiotemporal evolution of two chemical species parameterized by diffusion coefficients $D_A, D_B$, reaction strength $\delta$, feed rate $F$ for species $A$, and kill rate $k$ for species $B$.

**Training Data Generation.** We construct training data using two types of operators. First, *diffusion-kill operators* use fixed diffusion coefficients $D_A = 2 \times 10^{-5}$, $D_B = 1 \times 10^{-5}$, disabled reaction terms ($\delta = 0$, $F = 0$), and kill rates $k$ spanning 20 values in $\{0.051, 0.052, \ldots, 0.070\}$. Second, *pure reaction operators* disable diffusion ($D_A = D_B = 0$), set unit reaction strength ($\delta = 1$), zero kill rate ($k = 0$), and vary feed rates $F$ across 20 values in $\{5, 10, \ldots, 100\} \times 10^{-3}$.

The spatial domain employs a $128 \times 128$ grid with periodic boundary conditions. We generate 512 trajectories per parameter configuration, simulating 50 seconds and retaining 50 temporal snapshots using the solver from Ohana et al. (2024).

**Initial Conditions.** To ensure fair evaluation of dynamics identification and extrapolation capabilities, we address the distinct field characteristics produced by reaction versus diffusion dynamics. We begin with clustered Gaussian initial conditions, then evolve them for a random duration between 0 and 100 seconds using the full reaction-diffusion dynamics. The resulting evolved states serve as initial conditions for generating the isolated reaction and diffusion training trajectories. This procedure mitigates potential frequency bias across all methods and enables the assessment of operator learning rather than initial condition adaptation.

### A.4. Euler, Diffusion, and Navier–Stokes Equations

All data consist of trajectories of the two-dimensional vorticity field $\omega(t, x, y)$ on a periodic square domain $[0, 2\pi]^2$.

**Euler equations (inviscid).** The 2D incompressible Euler equations are solved in vorticity form:

$$\partial_t \omega + \mathbf{u} \cdot \nabla \omega = 0, \qquad \mathbf{u} = (-\partial_y \psi, \ \partial_x \psi), \qquad -\Delta \psi = \omega. \tag{7}$$

**Diffusion equation.** For the purely dissipative components, we also generate trajectories of the heat equation

$$\partial_t \omega = \nu \Delta \omega, \tag{8}$$

where $\nu > 0$ denotes the diffusion coefficient.

**Navier–Stokes equations.** We consider the two-dimensional incompressible Navier–Stokes equations as the viscous extension of the Euler equations, also written in vorticity form:

$$\partial_t \omega + \mathbf{u} \cdot \nabla \omega = \nu \Delta \omega, \qquad \mathbf{u} = (-\partial_y \psi, \ \partial_x \psi), \qquad -\Delta \psi = \omega. \tag{9}$$

**Numerical Discretization.** All simulations are performed using a Fourier pseudo-spectral method on a uniform $512 \times 512$ grid. Spatial derivatives are computed exactly in Fourier space, while nonlinear terms are evaluated in physical space.

For Euler and Navier–Stokes simulations, the nonlinear advection term is dealiased using the standard 3/2-rule padding. Time integration is carried out using an implicit midpoint scheme, solved via fixed-point iterations in spectral space. This method improves long-time stability and preserves invariants in the inviscid limit. The timestep is chosen adaptively according to a CFL condition, with an additional diffusion stability constraint when $\nu > 0$.

The diffusion equation is solved exactly in Fourier space using the closed-form solution

$$\hat{\omega}(t, k) = \hat{\omega}(0, k) \exp\left(-\nu |k|^2 t\right), \tag{10}$$

which introduces no time-discretization error beyond floating-point precision.

**Initial Conditions.** Initial vorticity fields are smooth and periodic. Euler initial conditions are generated as random low-frequency Fourier-mode mixtures of the form

$$\omega_0(x, y) = \sum_{n, m = 1}^{N_m} a_{nm} \sin\left(\tfrac{2\pi n}{L} x + \phi_{nm}\right) \sin\left(\tfrac{2\pi m}{L} y + \psi_{nm}\right), \tag{11}$$

where amplitudes $a_{nm}$ are Gaussian with variance proportional to $10 \, (n + m)^{-1}$, and phases $\phi_{nm}, \psi_{nm}$ are sampled uniformly from $[0, 2\pi]$. This construction ensures spatial smoothness and exact periodicity. We use $N_m = 5$ modes for all smooth initial conditions.

Diffusion initial conditions are obtained by sampling random intermediate snapshots (uniformly in time) from Euler trajectories. This yields physically realistic initial states containing coherent vortical structures and a broad range of spatial scales.

**Time Horizon and Downsampling.** Each trajectory is evolved over a fixed time horizon $T = 4.0$ and recorded at $n_{\text{snap}} = 50$ uniformly spaced time points.

For storage and learning, trajectories are downsampled to a lower output resolution (e.g., $256 \times 256$) using spectral truncation. Fourier modes outside the target bandwidth are removed, and the field is reconstructed via inverse FFT. This procedure preserves large-scale dynamics and avoids aliasing artifacts.

**Diffusion Viscosities** Diffusion trajectories for both the diffusion and Navier–Stokes datasets are generated for viscosities $\nu$ logarithmically spaced in the interval $[10^{-4}, 10^{-2}]$. The dataset is balanced such that an equal number of trajectories is generated for each viscosity value.

## B. Implementation Details

### B.1. DISCO implementation

**Hyperparameters** We use the recommended default configuration from (Morel et al., 2025) with targeted modifications for our experiments. For the transformer encoder, we employ a hidden dimension of 128, patch sizes of 8 in 1D and $8 \times 8$ in 2D, and 4 encoder blocks with 4 attention heads each using relative position bias. We introduce a bottleneck projection layer that reduces the 128-dimensional transformer output to $C$ channels, where $C = \{2, 3, 2\}$ for advection-diffusion, combined-equation, and reaction-diffusion respectively. This bottleneck layer is inserted before DISCO's original MLP decoder, representing a minimal architectural change that improves generalization across initial conditions.

For the neural ODE component, we select the RK4 solver with problem-specific integration time spans: $dt = \{0.1, 0.016, 1\}$ for advection-diffusion, combined-equation, and reaction-diffusion respectively. We apply periodic boundary conditions and configure the operator network with 8 base channels and a $2\times$ bottleneck multiplier for efficient ODE parameter prediction from transformer representations.

We train models for 300,000 iterations on advection-diffusion and combined-equation tasks, and 100,000 iterations for reaction-diffusion. We use the AdamW optimizer with a base learning rate of $3 \times 10^{-4}$, cosine annealing scheduler, and weight decay of $1 \times 10^{-4}$.

### B.2. Training recipe

The original DISCO training procedure uses the operator to predict the frame immediately following the input encoder sequence. We found that increasing input diversity to the operator network produces more robust operators that generalize across different initial conditions.

We therefore propose an alternative training strategy based on contextual learning. For advection-diffusion equations, we sample two trajectories that follow identical dynamics: we encode trajectory 1 with the hypernetwork to obtain an operator, then apply this operator to predict the next timestep of trajectory 2. This in-context approach draws inspiration from Serrano et al. (2025).

For Combined-equation and Gray–Scott systems, we adopt an environment-based training paradigm to enable fair comparison with GEPS (Koupaï et al., 2024). We assume knowledge of which trajectories belong to the same environment and implement a codebook updated via exponential moving average following (Oord et al., 2017). During training, we randomly select either the encoder-derived code (50% probability) or the corresponding environment code from the codebook (50% probability), ensuring the encoder learns meaningful representations while maintaining environment consistency.

### B.3. Operator splitting with $m$ operators

For multiple operators $f_{i_1} + f_{i_2} + \cdots + f_{i_m}$, we generalize the composition by sequentially composing the individual operator flows. The Lie splitting for $m$ operators is given by

$$\hat{u}^{L+1} = f_{i_m}^{\Delta t} \circ f_{i_{m-1}}^{\Delta t} \circ \cdots \circ f_{i_1}^{\Delta t} \left( u^L \right),$$

yielding a first-order approximation of the full evolution operator.

To obtain second-order accuracy, we employ a symmetric Strang-type splitting for multiple operators:

$$\hat{u}^{L+1} = f_{i_1}^{\Delta t/2} \circ f_{i_2}^{\Delta t/2} \circ \cdots \circ f_{i_{m-1}}^{\Delta t/2} \circ f_{i_m}^{\Delta t} \circ f_{i_{m-1}}^{\Delta t/2} \circ \cdots \circ f_{i_1}^{\Delta t/2} \left( u^L \right).$$

---

**Algorithm 1** Beam Search Operator Composition

---

**Require:** Test trajectory $u_{\text{test}}^{1:L}$, dictionary $\{f_1, \ldots, f_N\}$, beam width $B$, max iterations $M$, threshold $\tau$
**Ensure:** Best operator subset $S^*$
 1: Initialize: $\mathcal{B}_0 = $ top-$B$ operators ranked by $\mathcal{L}(\{f_i\})$
 2: **for** $m = 0$ to $M - 1$ **do**
 3:     Candidates $= \emptyset$
 4:     **for** each $S \in \mathcal{B}_m$ **do**
 5:         **for** each $f_j \in \{f_1, \ldots, f_N\} \setminus S$ **do**
 6:             Add $S \cup \{f_j\}$ to Candidates
 7:         **end for**
 8:     **end for**
 9:     $\mathcal{B}_{m+1} = $ top-$B$ from Candidates ranked by $\mathcal{L}(\cdot)$
10:     **if** relative improvement $< \tau$ or $m = M - 1$ **then**
11:         **break**
12:     **end if**
13: **end for**
14: **return** $\arg\min_{S \in \mathcal{B}_m} \mathcal{L}(S)$

---

**Algorithm 2** Uniform Operator Composition Search

---

**Require:** Test trajectory $u_{\text{test}}^{1:L}$, operator dictionary $\{f_1, \ldots, f_N\}$, number of trials $N_{\text{trials}}$, maximum composition length $M$
**Ensure:** Best operator subset $S^*$
 1: Initialize best operator subset $S^* = \{\arg\min_{f_i \in \{f_1, \ldots, f_N\}} \mathcal{L}(\{f_i\})\}$
 2: Initialize best loss $\mathcal{L}^* = \mathcal{L}(S^*)$
 3: **for** $i = 1$ to $N_{\text{trials}}$ **do**
 4:     Sample composition length $m \sim \text{Uniform}(\{1, 2, \ldots, M\})$
 5:     Sample operator subset $S_i \sim \text{Uniform}(\text{subsets of } \{f_1, \ldots, f_N\} \text{ with size } m)$
 6:     Compute loss $\mathcal{L}_i = \mathcal{L}(S_i)$
 7:     **if** $\mathcal{L}_i < \mathcal{L}^*$ **then**
 8:         $S^* = S_i$
 9:         $\mathcal{L}^* = \mathcal{L}_i$
10:     **end if**
11: **end for**
12: **return** $S^*$

---

This palindromic composition preserves the symmetry required for second-order accuracy and extends classical Strang splitting to multiple operators (Strang, 1968; Marchuk, 1990).

### B.4. Baselines

**FNO**  We employ the FNO1D and FNO2D implementations, and treat the temporal dimension as additional input channel to the system, i.e. the temporal context of length $L = 16$ is fed into the channel dimension of the model. Training uses the relative $L^2$ loss with AdamW ($\beta_1$=0.9, $\beta_2$=0.999, weight decay $10^{-4}$) and a cosine schedule with 5% linear warmup. All variants use hidden width 128, $L$=4 Fourier blocks, and a spectral truncation of 16 modes per dimension (i.e., 16 for FNO1D and 16×16 for FNO2D), yielding ~2.2M parameters in 1D and ~67M in 2D. Per-benchmark training budgets match those of MPP for direct comparability: 100k steps with batch size 64 and learning rate $5\times10^{-4}$ for advection–diffusion, 100k steps with batch size 64 and learning rate $3\times10^{-4}$ for the combined equation, 300k steps with batch size 16 and learning rate $2\times10^{-4}$ for Gray–Scott, and 150k steps with batch size 8 and learning rate $2\times10^{-4}$ for Navier–Stokes. Each run uses a single NVIDIA A100 GPU.

**MPP**  We use the recommended default hyperparameters with periodic boundary conditions and 6 encoder blocks, employing a hidden dimension of 384 for 2D experiments, and train for 100,000 iterations using the AdamW optimizer with a learning rate of $5 \times 10^{-4}$ and batch size of 64.

**Zebra**  We adopt the recommended default configuration, using 64, 32, and 256 tokens respectively to encode each frame for advection-diffusion, combined-equation, and reaction-diffusion tasks. We train without in-context examples, employing maximum history lengths of 50, 66, and 32 frames for advection-diffusion, combined-equation, and reaction-diffusion respectively. At inference, we sample the next token from the multinomial distribution with a temperature of 0.1 to reduce the variance.

**GEPS**  We use the CNN1D and CNN2D implementations from the original codebase, training for 100,000 steps with the AdamW optimizer and cosine learning rate scheduling. Since GEPS requires environment information during training, we provide labels indicating which trajectories belong to the same environment. At inference, we address rollout instabilities by performing multiple optimization runs (100, 500, and 2000 gradient steps) and report the best test set performance across these attempts.

## C. Additional Experimental results

### C.1. Splitting Accuracy and Individual Operator Quality

We investigate how approximation error in individual learned neural operators influences the accuracy of their composition. Although operator splitting is well studied in classical numerical analysis, the extent to which splitting interacts with learned operator error is not obvious *a priori* and therefore benefits from empirical evaluation.

**Experimental setup.**  We consider a 1D heat–dispersion system and train separate neural operators for each component over multiple pretraining epochs, producing operators with varying levels of accuracy. At evaluation time, we compose these learned components using Strang splitting, and report both single-step prediction error and long-horizon rollout error (250 steps).

**Results.**  Overall, the accuracy of the composed system largely tracks the least accurate constituent operator. In particular, when one component exhibits substantially higher error than the other, the splitting error is typically governed by that weaker operator, even when the second component is already highly accurate. As both operators improve with continued pretraining, the composed error decreases consistently and proportionally. We note one exception at the highest-accuracy setting (last row of Table 3), where the composed error is slightly lower than the larger of the two individual operator errors, suggesting that composition can occasionally be marginally better than a strict "worst-operator" bound. Nonetheless, the overall trend remains stable, with no evidence of unexpected degradation from composition.

**Interpretation**  These results demonstrate a clear "weakest-link" behavior in learned operator splitting: improvements to an already accurate component yield limited overall benefit unless the less accurate operator is also improved. At the same time, the monotonic reduction in error across training stages suggests that Strang splitting does not introduce pathological

*Table 3.* Operator splitting accuracy vs. individual operator quality. Composition of learned heat diffusion and dispersion operators across pretraining epochs. *Heat Err* and *Dispersion Err* report the next-step NRMSE of each individual learned operator; *Split Next-Step* reports the next-step NRMSE of the Strang composition; *Split Rollout* reports NRMSE averaged over a 250-step autoregressive rollout. The composed system's error is dominated by the least accurate individual operator.

| Epoch | Heat Err | Dispersion Err | Split Next-Step | Split Rollout |
|---|---|---|---|---|
| 50 | 5.39e-05 | 1.47e-03 | 1.24e-03 | 1.22e-01 |
| 150 | 3.45e-05 | 1.02e-03 | 8.46e-04 | 1.02e-01 |
| 250 | 2.08e-05 | 6.18e-04 | 6.55e-04 | 9.90e-02 |
| 350 | 5.84e-06 | 1.52e-04 | 1.93e-04 | 2.73e-02 |
| 500 | 2.59e-06 | 8.95e-05 | 8.26e-05 | 8.85e-03 |

interactions between learned operators. Overall, this supports the practical value of modular training, while highlighting that performance gains in composed systems are driven primarily by progress on the most challenging component.

### C.2. Robustness to Unseen Perturbation Terms

We stress-test the splitting framework when test trajectories contain a term that is *not* representable by any operator in the training dictionary. Recall that the advection–diffusion training data consists exclusively of pure-advection or pure-diffusion trajectories. We introduce a nonlinear Burgers-like perturbation at test time:

$$\frac{\partial u}{\partial t} + v\,\frac{\partial u}{\partial x} + \varepsilon \cdot u\,\frac{\partial u}{\partial x} = D\,\frac{\partial^2 u}{\partial x^2},$$

where $\varepsilon = 0$ recovers the standard OOD advection+diffusion test case used in the main paper, and $\varepsilon > 0$ introduces a nonlinear advection term that lies strictly outside the span of dictionary operators. We sweep $\varepsilon \in \{0, 0.01, 0.05, 0.10, 0.25, 0.50, 1.00\}$ at fixed $v = 0.5$ and $D = 0.3$, using 128 test trajectories per $\varepsilon$ and rolling out for 84 steps.

*Table 4.* Robustness of the splitting framework to an unseen Burgers-like perturbation. *Direct NRMSE* reports the direct DISCO prediction; *Beam NRMSE* reports our test-time beam-search prediction. *Est. $v$* and *Est. $D$* are the recovered PDE coefficients; the true values are $v = 0.5$ and $D = 0.3$.

| $\varepsilon$ | Direct NRMSE | Beam NRMSE | Est. $v$ | Est. $D$ |
|---|---|---|---|---|
| 0.00 | 0.319 | **0.055** | 0.495 | 0.316 |
| 0.01 | 0.323 | **0.064** | 0.507 | 0.328 |
| 0.05 | 0.355 | **0.061** | 0.518 | 0.315 |
| 0.10 | 0.417 | **0.089** | 0.511 | 0.279 |
| 0.25 | 0.704 | **0.184** | 0.502 | 0.276 |
| 0.50 | 0.843 | **0.414** | 0.496 | 0.125 |
| 1.00 | 0.817 | **0.797** | 0.496 | 0.048 |

**Results.** Performance degrades gracefully: for $\varepsilon \leq 0.05$ accuracy is essentially unaffected, and beam search strictly outperforms direct prediction at every $\varepsilon$. The advection coefficient $v$ is recovered correctly across the entire range, even when the dynamics are no longer exactly decomposable into operators seen during training. The diffusion estimate $\hat{D}$ degrades at large $\varepsilon$, where the splitting framework increasingly absorbs the unseen nonlinear term into the diffusion component.

### C.3. Stress Test: Mixed-Physics Training

The main-paper setting trains operators exclusively on separated-physics trajectories. To test whether the splitting framework remains effective when training trajectories already entangle multiple physical mechanisms, we retrain DISCO on advection–diffusion data under two mixed-training regimes:

- **Mixed 0.5**: 50% coupled advection+diffusion trajectories, 25% pure advection, 25% pure diffusion. The resulting dictionary contains a mixture of pure and coupled operators.

- **Mixed 1.0**: 100% coupled trajectories. The dictionary contains only coupled operators; no pure-advection or pure-diffusion operator exists.

We evaluate on joint parameter extrapolation, drawing both coefficients from $[0.01, 2]$, and report results for two dictionary sizes $N \in \{64, 256\}$.

*Table 5.* Beam-search performance under mixed-physics training. Even when no pure-physics operator exists in the dictionary (Mixed 1.0), beam search recovers meaningful gains over direct prediction once the dictionary is large enough to span a diverse set of compositions.

| Model | $N$ | Direct | Beam | Improvement |
|---|---|---|---|---|
| Mixed 0.5 | 64 | 0.098 | 0.097 | 0% |
| Mixed 0.5 | 256 | 0.097 | **0.057** | 41% |
| Mixed 1.0 | 64 | 0.313 | 0.267 | 15% |
| Mixed 1.0 | 256 | 0.278 | **0.190** | 32% |

**Results.** Beam-search gains scale with dictionary size. At $N = 64$ under Mixed 0.5, there are not enough diverse compositions for the search to outperform direct prediction; at $N = 256$, the same training regime yields a 41% improvement. Under the most adversarial Mixed 1.0 setting, where the dictionary contains *only* coupled operators and no pure-physics building blocks, beam search still delivers a 32% improvement at $N = 256$. This suggests that the splitting framework does not strictly require single-physics training trajectories: provided the dictionary is sufficiently large and diverse, the search can recover useful compositions even from entangled operators. Single-physics training, as used in the main paper, remains the cleanest setup for interpretability and parameter recovery, but the method itself is not strictly dependent on it.

### C.4. Ablation: In-Context Conditioning and Codebook

The dictionary construction described in Section 4.1 is implemented with either in-context conditioning (used for advection-equation), or a dynamics codebook (e.g. used for 2D Navier–Stokes). See Section B.2 for a description of these two elements. To isolate their contribution from the test-time search, we retrain DISCO under two ablated regimes—without in-context learning on the advection-diffusion task, and without the codebook on Navier–Stokes—and rerun beam search at test time.

*Table 6.* Effect of the DISCO training recipe on beam-search performance. *Paper* corresponds to the configuration used in the main paper (in-context for 1D, codebook for 2D); *No-context / No-codebook* removes the respective component. Both ablations use identical test-time search.

| Setting | Paper | No-context / No-codebook |
|---|---|---|
| Advection + Diffusion | **0.015** | 0.026 |
| Advection extrapolation | **0.052** | 0.406 |
| Diffusion extrapolation | **0.002** | 0.012 |
| Navier–Stokes | **0.066** | 0.563 |

**Results.** Either training-recipe choice—in-context conditioning (1D) or the codebook (2D)—contributes to downstream search effectiveness, but the gap widens under more complex OOD extrapolation: removing in-context conditioning causes the advection-extrapolation error to grow nearly $8\times$, and removing the codebook on Navier–Stokes degrades performance by an order of magnitude. Without the improved training recipe, we hypothesize that the learned operators specialize to the initial-condition distribution of their training trajectories and transfer poorly to unseen configurations, even under our test-time search. This indicates that the modified training recipe and the test-time search procedure are *complementary*: neither alone is sufficient for far-OOD generalization.

### C.5. Sensitivity to Dictionary Size $N$

The main paper uses dictionary sizes $N \in \{17, 40, 96, 256\}$ depending on the benchmark, obtained by selecting one representative operator per training environment. To verify that performance is not overly sensitive to this choice, we sweep $N \in \{16, 32, 64, 256\}$ on the advection+diffusion benchmark while keeping all other hyperparameters fixed.

**Results.** Performance is stable even at the smallest $N = 16$, and saturates around $N = 64$. The marginal gain from $N = 64$ to $N = 256$ is small ($0.017 \rightarrow 0.015$), confirming that the search is robust to the dictionary-subsampling strategy and that significantly smaller dictionaries can be used to reduce search cost without substantially degrading accuracy.

*Table 7.* Beam-search NRMSE on advection+diffusion as a function of dictionary size $N$. Performance is stable at small $N$ and saturates around $N = 64$.

| $N$ | Beam NRMSE |
|---|---|
| 16 | 0.037 |
| 32 | 0.035 |
| 64 | 0.017 |
| 256 (paper default) | **0.015** |

## C.6. Sensitivity to Fitting-Window Length $L$

The main paper uses $L = 16$ snapshots as the fitting window over which the search objective is computed. We sweep $L \in \{2, 4, 8\}$ on the advection+diffusion benchmark to evaluate sensitivity to this choice.

*Table 8.* Beam-search NRMSE on advection+diffusion as a function of the fitting-window length $L$.

| $L$ | Beam NRMSE |
|---|---|
| 2 | **0.011** |
| 4 | 0.013 |
| 8 | 0.013 |
| 16 (paper default) | $\sim$0.015 |

**Results.** The method is essentially insensitive to $L$ on advection+diffusion, and a much shorter fitting window ($L = 2$) suffices to recover the same accuracy as the default $L = 16$. Since the per-candidate cost of the search scales linearly in $L$, this opens a substantial speedup for tasks where the dynamics can be identified from a few snapshots. We expect this robustness to depend on the dataset—in particular, more chaotic or slowly-mixing dynamics may require larger $L$ for reliable identification.

## C.7. Wall-Clock Comparison with Gradient-Based Adaptation

We compare per-sample wall-clock time for our test-time beam search against GEPS gradient-based fine-tuning, the main adaptive baseline in our experiments. All measurements are on a single NVIDIA A100. For GEPS we report three checkpoints along the fine-tuning trajectory (100, 500, and 2000 steps) to characterize the accuracy–compute tradeoff.

*Table 9.* Per-sample NRMSE / wall-clock time on a single A100, for GEPS fine-tuning at 100/500/2000 steps versus our test-time beam search. GEPS diverges on both extrapolation tasks at 2000 steps; beam search remains stable on all three.

| Method | Steps | Adv+Diff | Adv. extrap. | Diff. extrap. |
|---|---|---|---|---|
| GEPS | 100 | 0.265 / $\sim$3.5 s | 1.129 / $\sim$3.5 s | 0.599 / $\sim$3.5 s |
| GEPS | 500 | 0.103 / $\sim$15 s | 0.847 / $\sim$15 s | 0.268 / $\sim$15 s |
| GEPS | 2000 | 0.034 / $\sim$57 s | Inf (diverged) | Inf (diverged) |
| Ours (Beam) | — | **0.015** / $\sim$21 s | **0.052** / $\sim$26 s | **0.002** / $\sim$26 s |

**Results.** On advection+diffusion, beam search is both $\sim 2.7\times$ faster and $\sim 2\times$ more accurate than GEPS fine-tuned for 2000 steps. On the harder extrapolation tasks, GEPS diverges entirely after extended fine-tuning, while beam search remains stable and achieves the lowest NRMSE of any method at comparable or lower wall-clock cost. The asymmetry is informative: adding compute via gradient steps actively hurts GEPS in far-OOD regimes, whereas adding compute via more search candidates consistently helps our method (see also Figure 5, which shows monotone improvement of beam search with search budget).

# D. Qualitative results

**Combined equation** We can see in Figure 6 , 7, 8, 9 that our test-time operator splitting strategy demonstrates remarkable capability in matching ground truth dynamics over extended rollouts, despite operating out-of-distribution and being trained solely for single-step prediction. The model maintains high fidelity predictions throughout the majority of the 100-step rollout, with some error accumulation becoming visible after approximately 70 timesteps, which is expected for such

long-horizon extrapolation tasks.

**Reaction diffusion**    We provide an augmented comparison of the dynamics seen during training with the second channel in Figure 10. We also show additional comparisons of predictions and ground truths in Figure 11, 12, 13.

**Navier–Stokes**    Figures 14, 15, 16, and 17 contrast direct predictions with our test-time strategy for OOD Navier–Stokes trajectories. The test-time strategy achieves lower error, and the qualitative improvement is especially noticeable in the higher-viscosity regime.

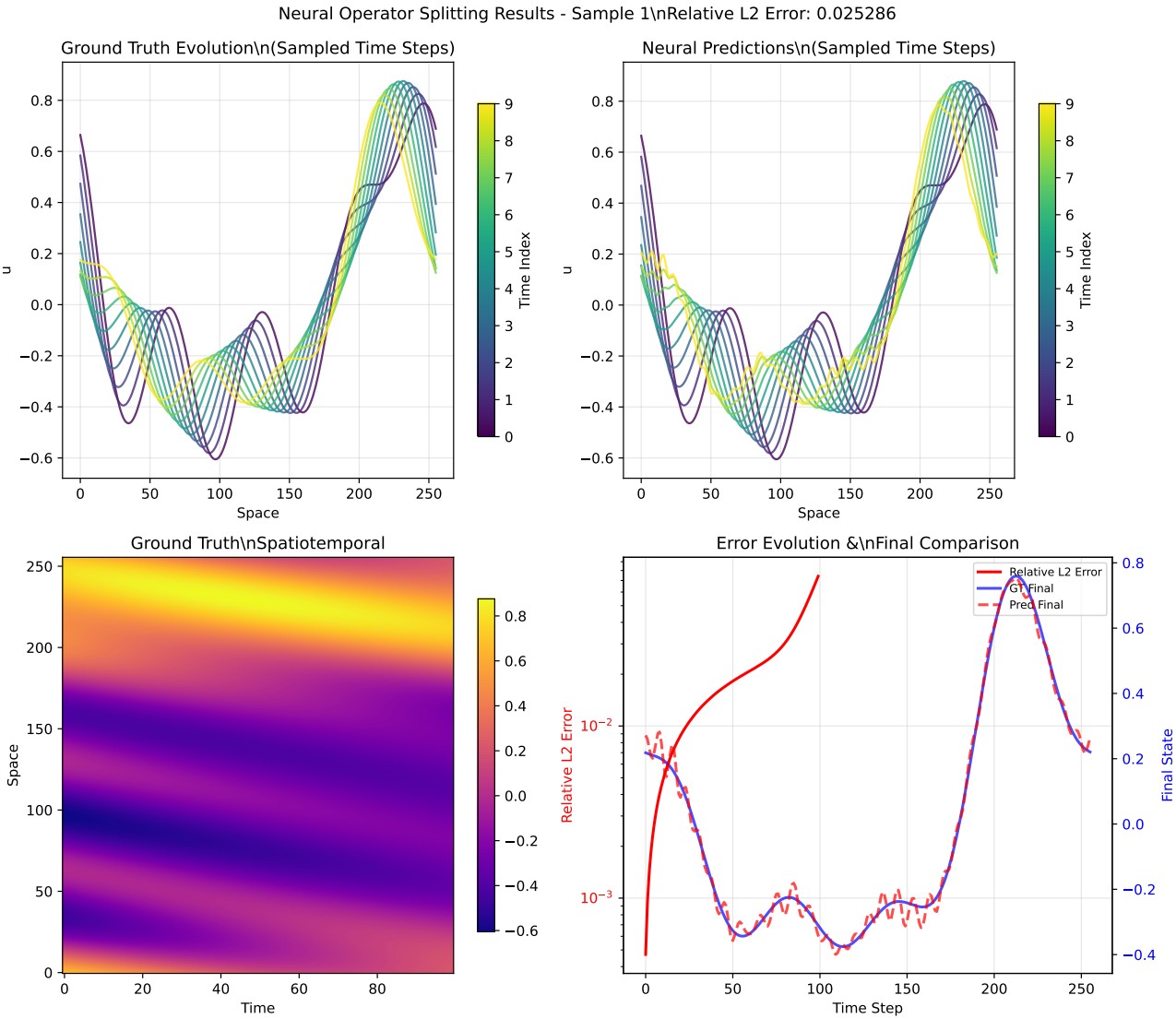

*Figure 6.* **OOD trajectory prediction on diffusion+dispersion equations.** We select operators using beam search and autoregressively unroll dynamics for 100 timesteps. The top panels show ground truth (left) and model predictions (right). The bottom panel displays error evolution throughout the rollout and compares the final prediction against ground truth.

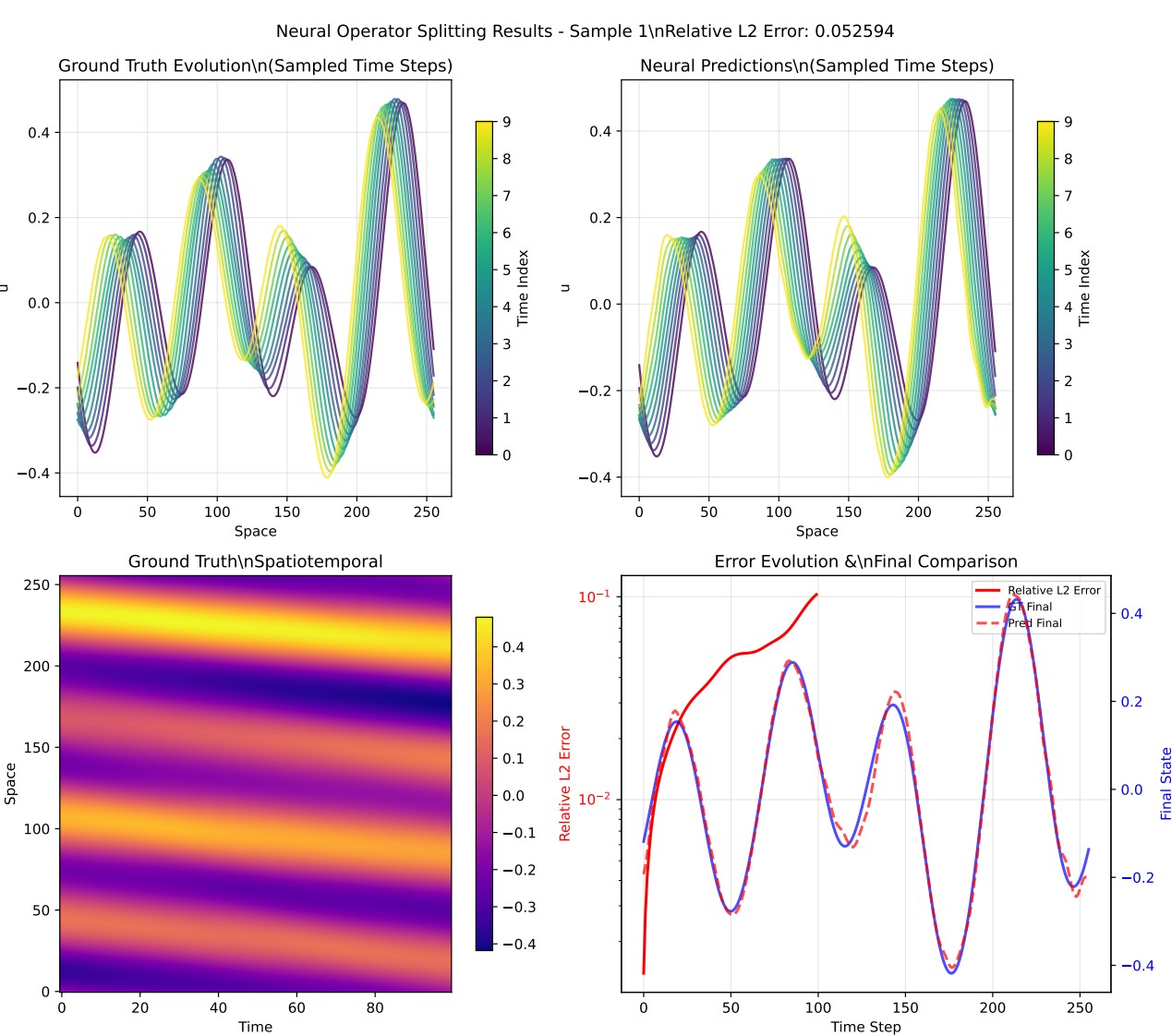

*Figure 7.* **OOD trajectory prediction on nonlinear advection+dispersion.** We select operators using beam search and autoregressively unroll dynamics for 100 timesteps. The top panels show ground truth (left) and model predictions (right). The bottom panel displays error evolution throughout the rollout and compares the final prediction against ground truth.

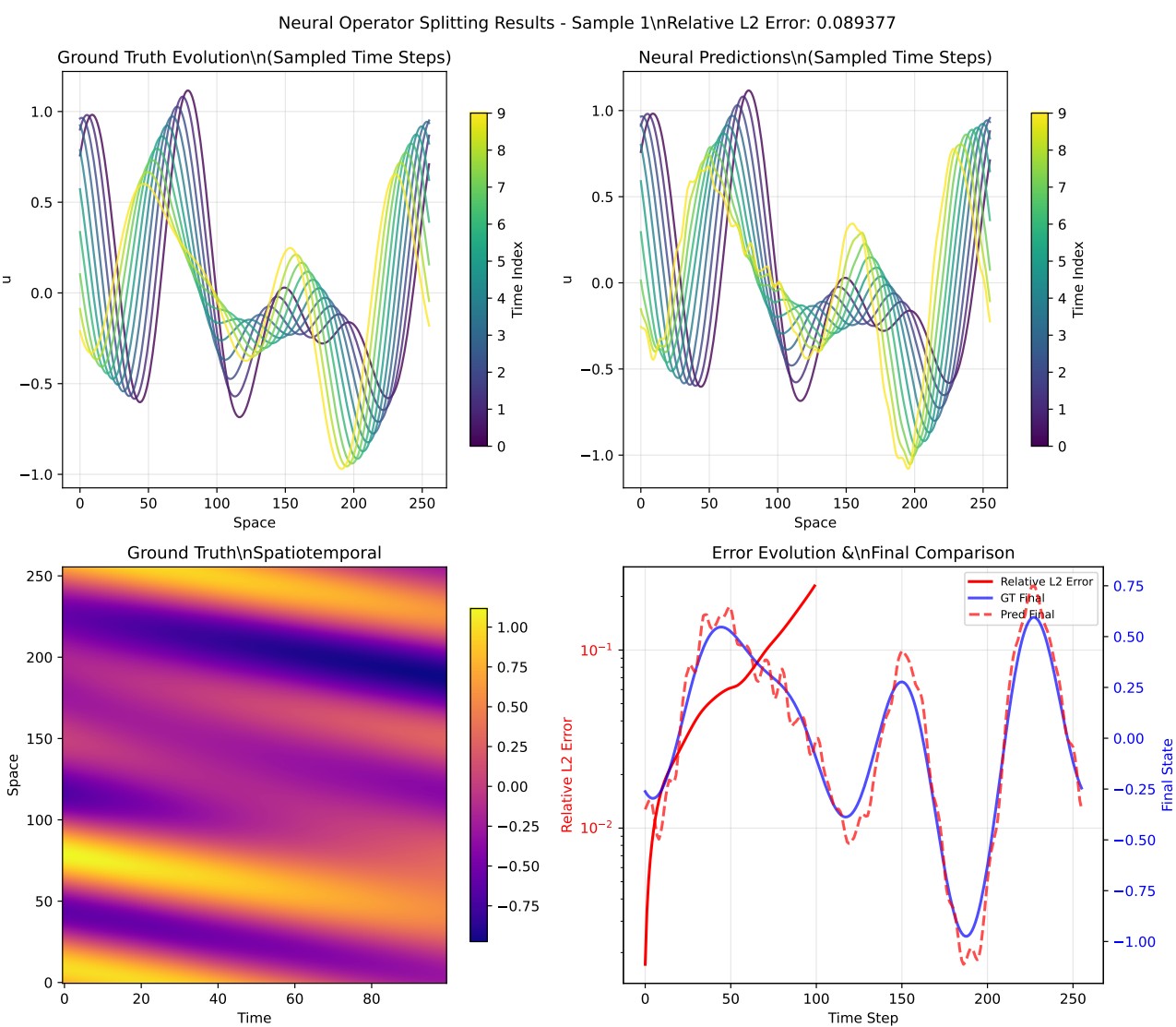

*Figure 8.* **OOD trajectory prediction on nonlinear advection+dispersion+diffusion.** We select operators using beam search and autoregressively unroll dynamics for 100 timesteps. The top panels show ground truth (left) and model predictions (right). The bottom panel displays error evolution throughout the rollout and compares the final prediction against ground truth.

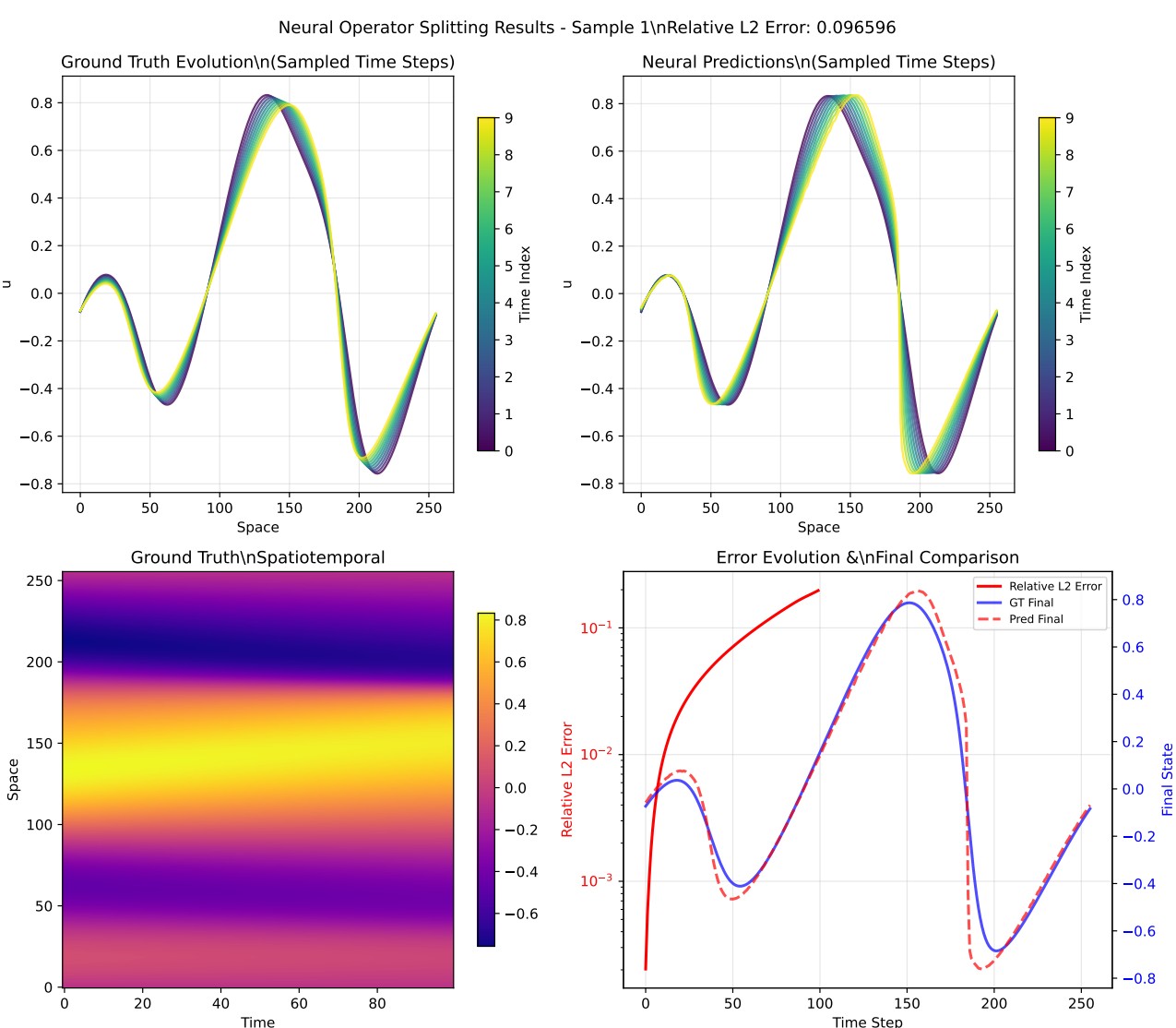

*Figure 9.* **OOD trajectory prediction on nonlinear advection+diffusion.** We select operators using beam search and autoregressively unroll dynamics for 100 timesteps. The top panels show ground truth (left) and model predictions (right). The bottom panel displays error evolution throughout the rollout and compares the final prediction against ground truth.

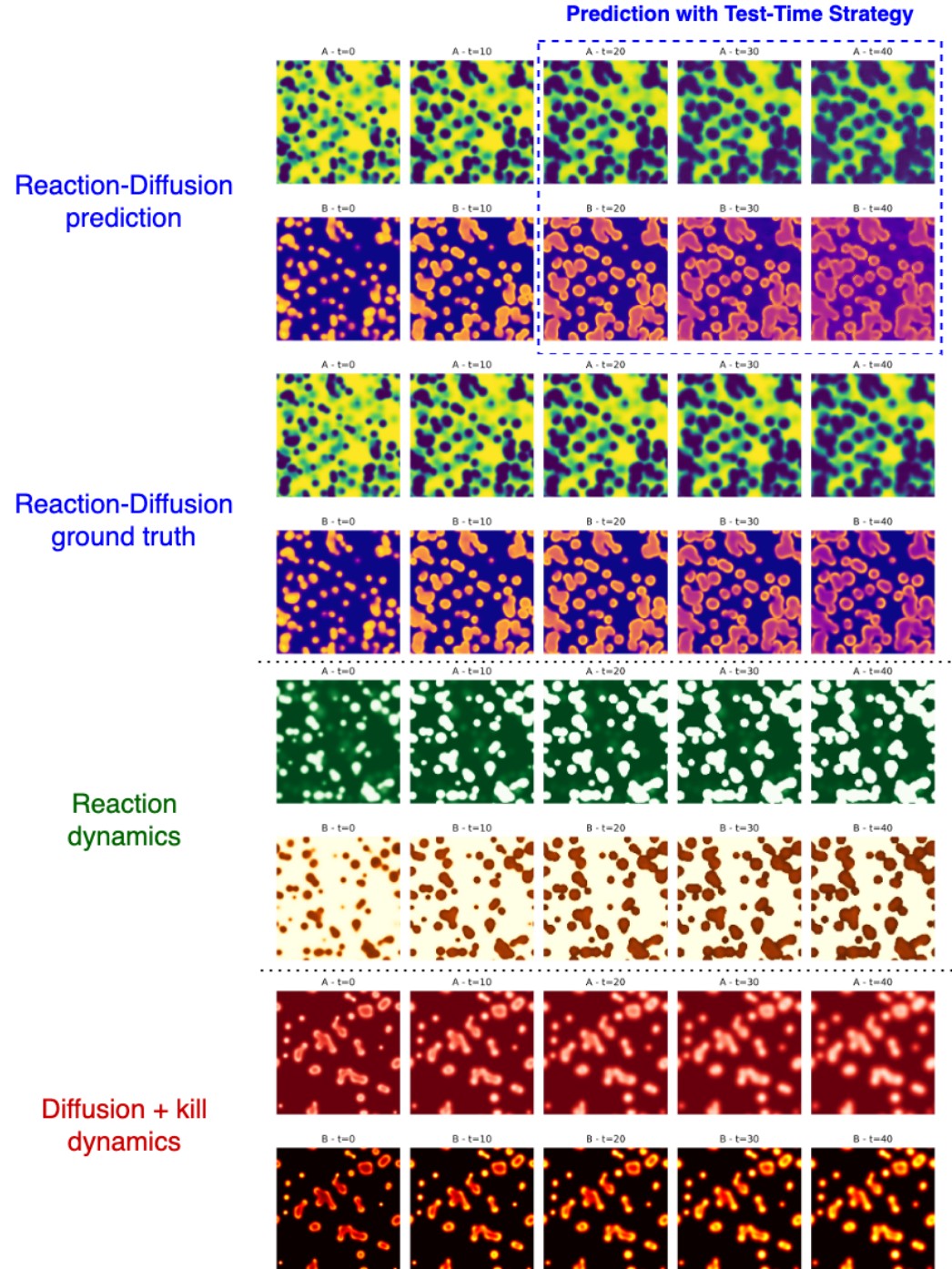

*Figure 10.* **OOD trajectory prediction on Gray–Scott equations.** Visualization of operator splitting decomposition for Gray–Scott reaction-diffusion dynamics. The top section compares our test-time strategy predictions (blue box) against ground truth for the full reaction-diffusion system, showing species A (yellow-green) and B (red-blue) concentrations. The bottom section displays the kind of dynamics seen during training: pure reaction terms (green/brown) and diffusion with kill terms (red/orange) for both species.

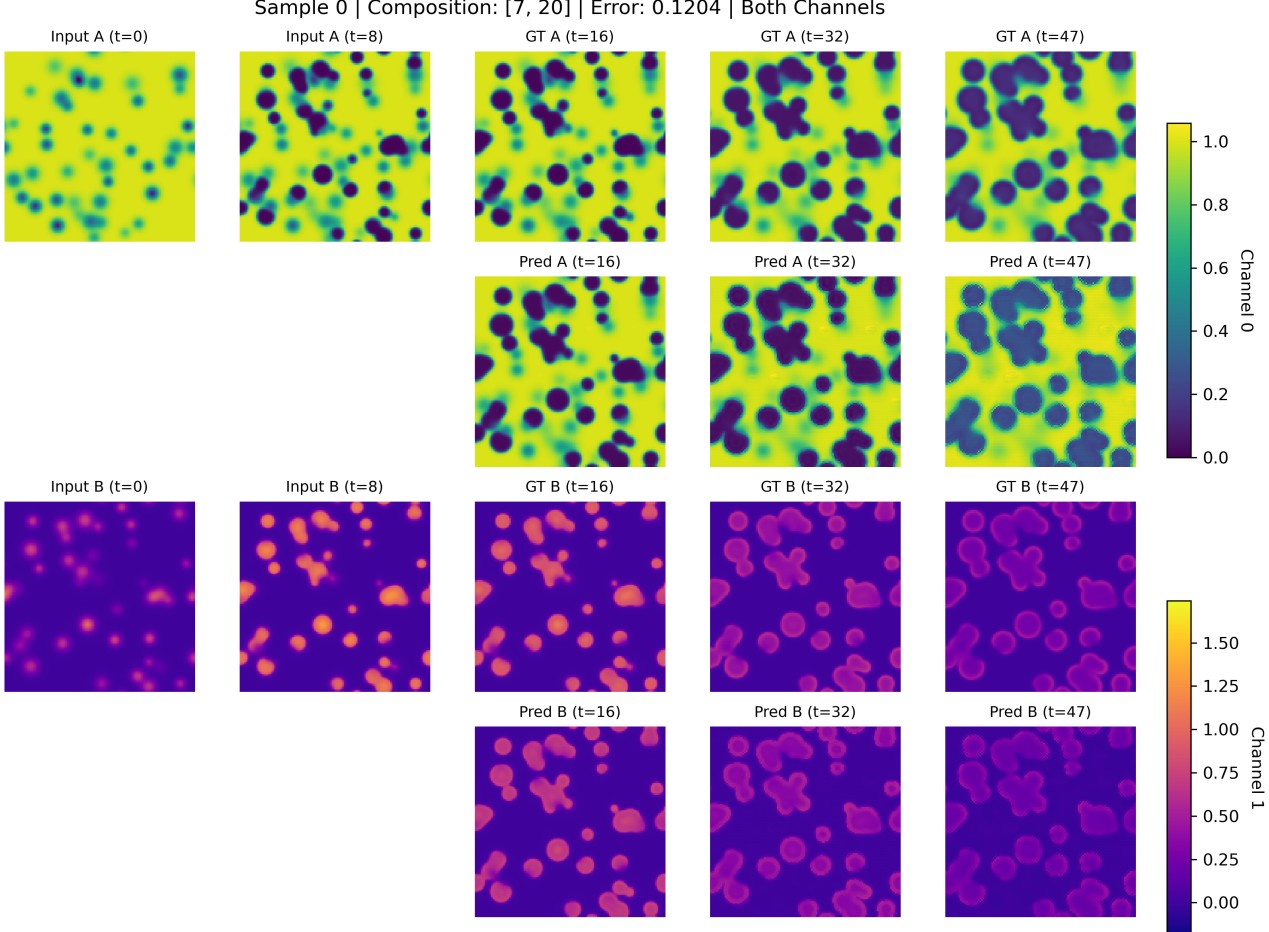

*Figure 11.* **OOD trajectory prediction on Gray–Scott equations.** The first two rows show ground truth (top) and predicted (second) concentrations for species $A$. The bottom two rows display ground truth (third) and predicted (bottom) concentrations for species $B$.

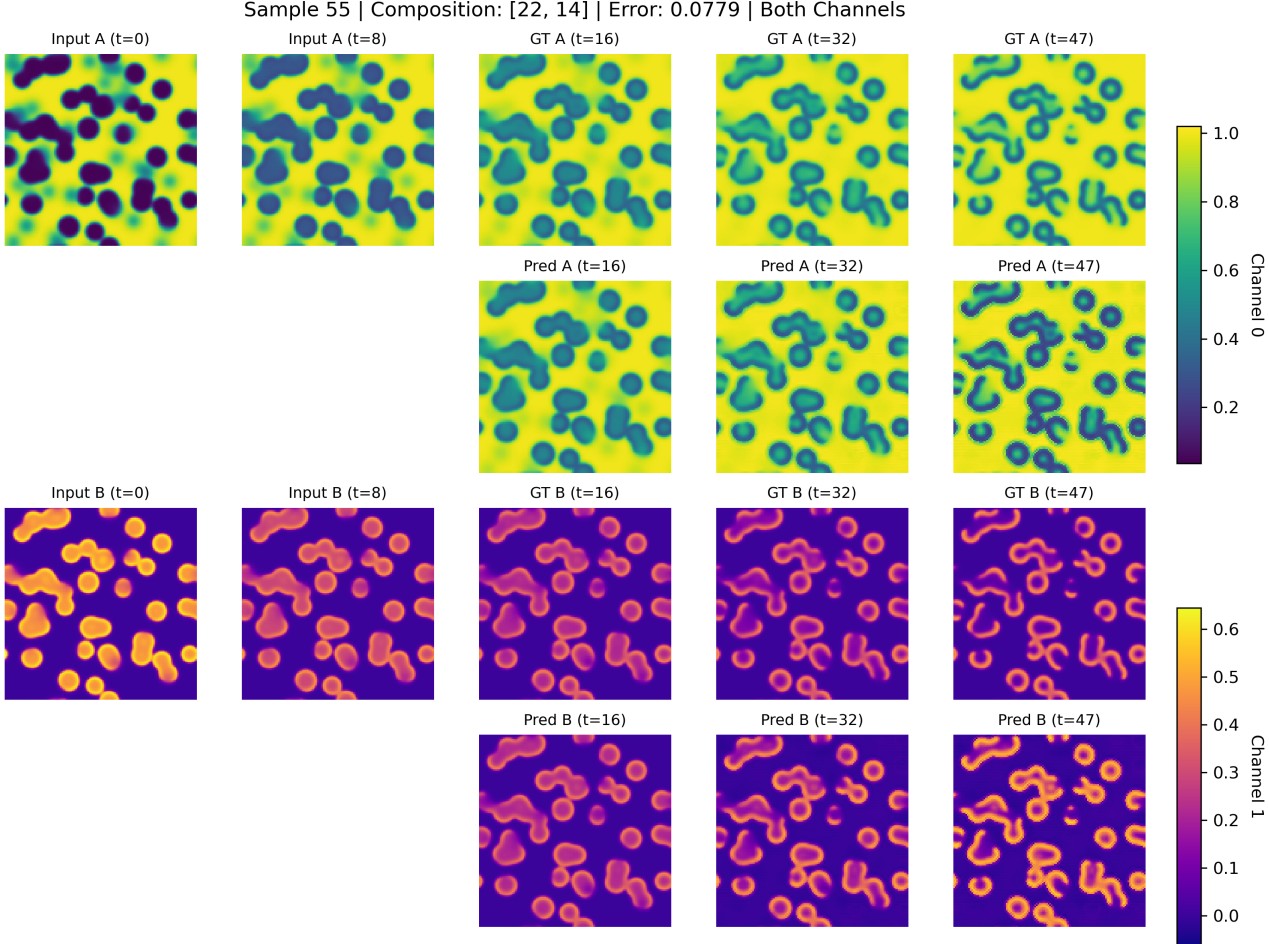

*Figure 12.* **OOD trajectory prediction on Gray–Scott equations.** The first two rows show ground truth (top) and predicted (second) concentrations for species $A$. The bottom two rows display ground truth (third) and predicted (bottom) concentrations for species $B$.

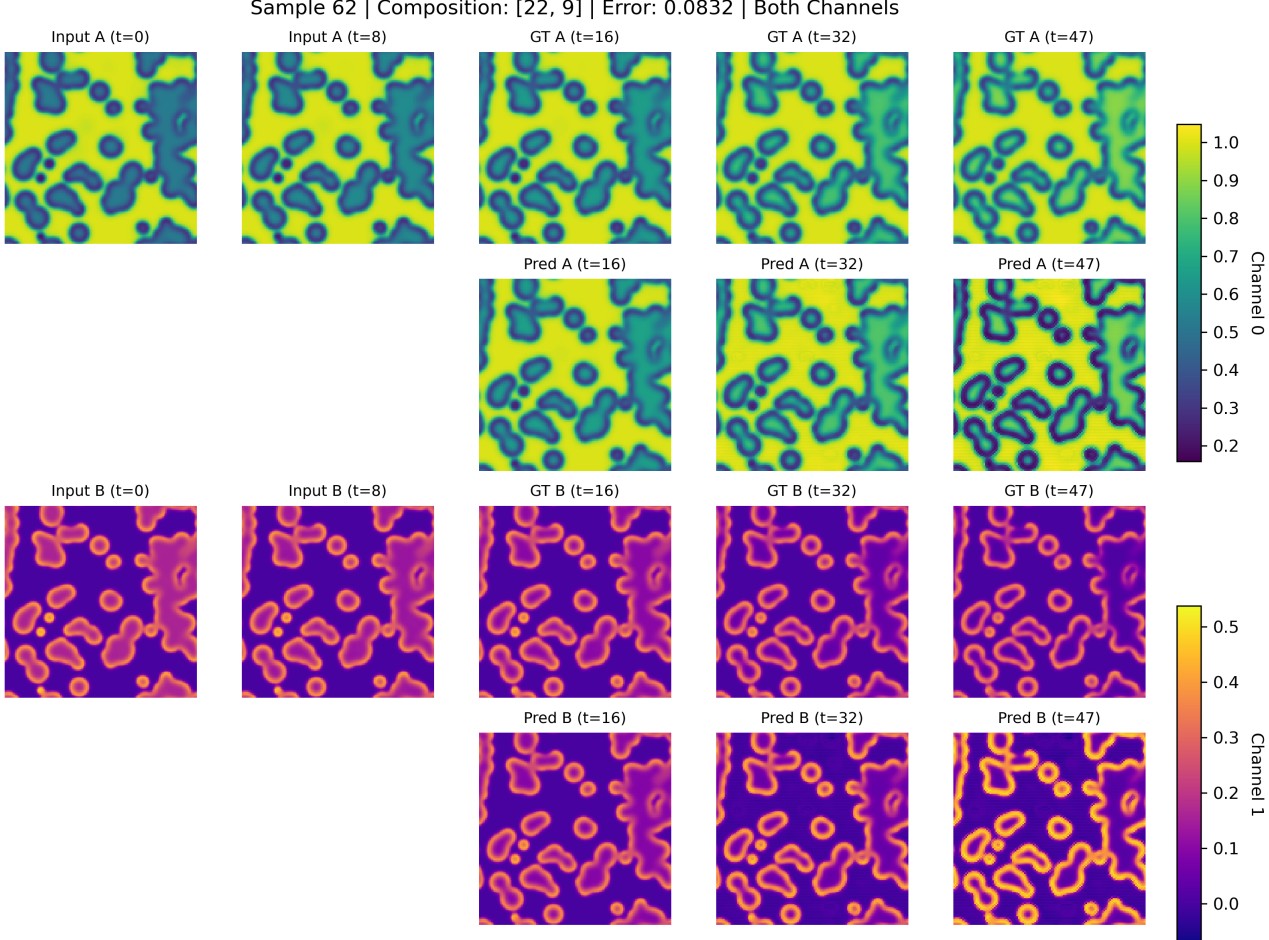

*Figure 13.* **OOD trajectory prediction on Gray–Scott equations.** The first two rows show ground truth (top) and predicted (second) concentrations for species $A$. The bottom two rows display ground truth (third) and predicted (bottom) concentrations for species $B$.

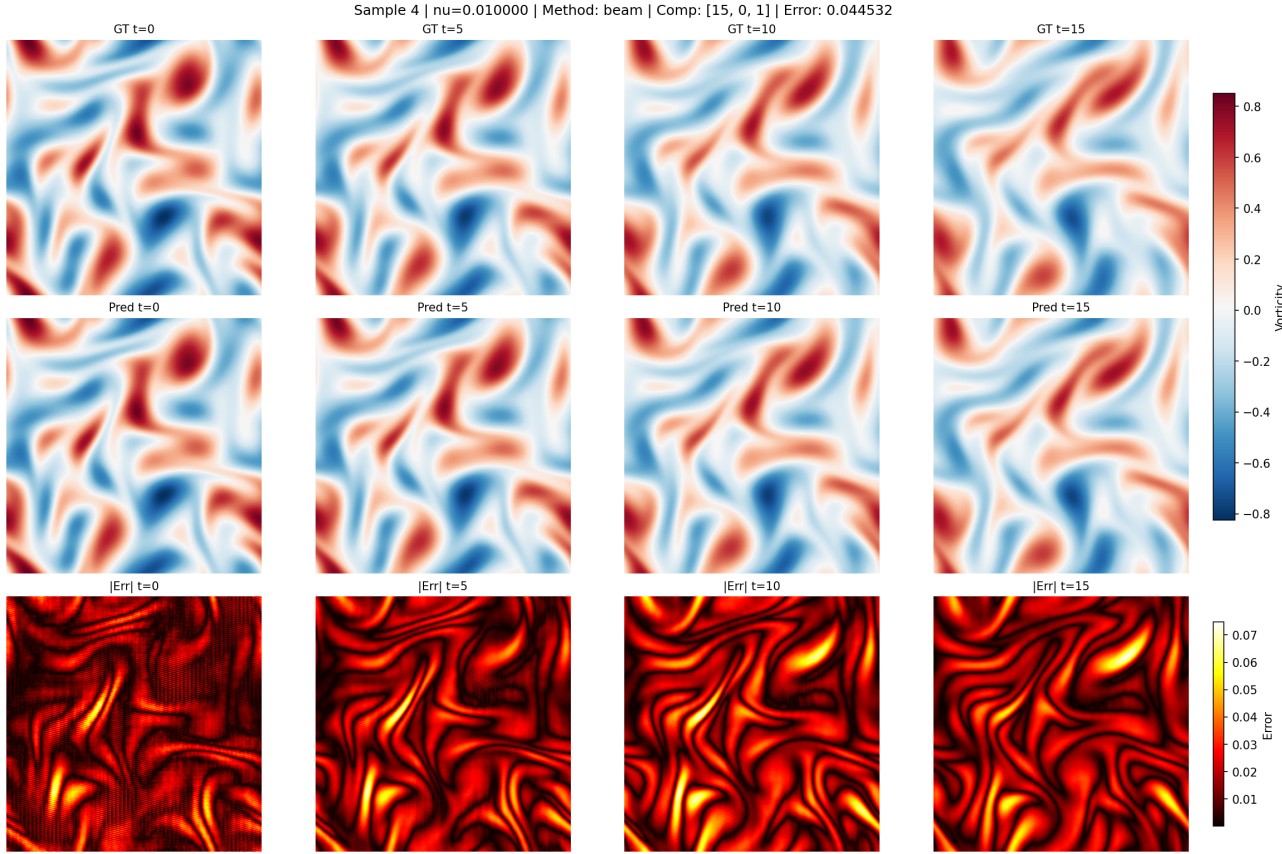

*Figure 14.* **OOD trajectory prediction on 2D Navier–Stokes.** Beam search composition with viscosity $\nu = 10^{-2}$.

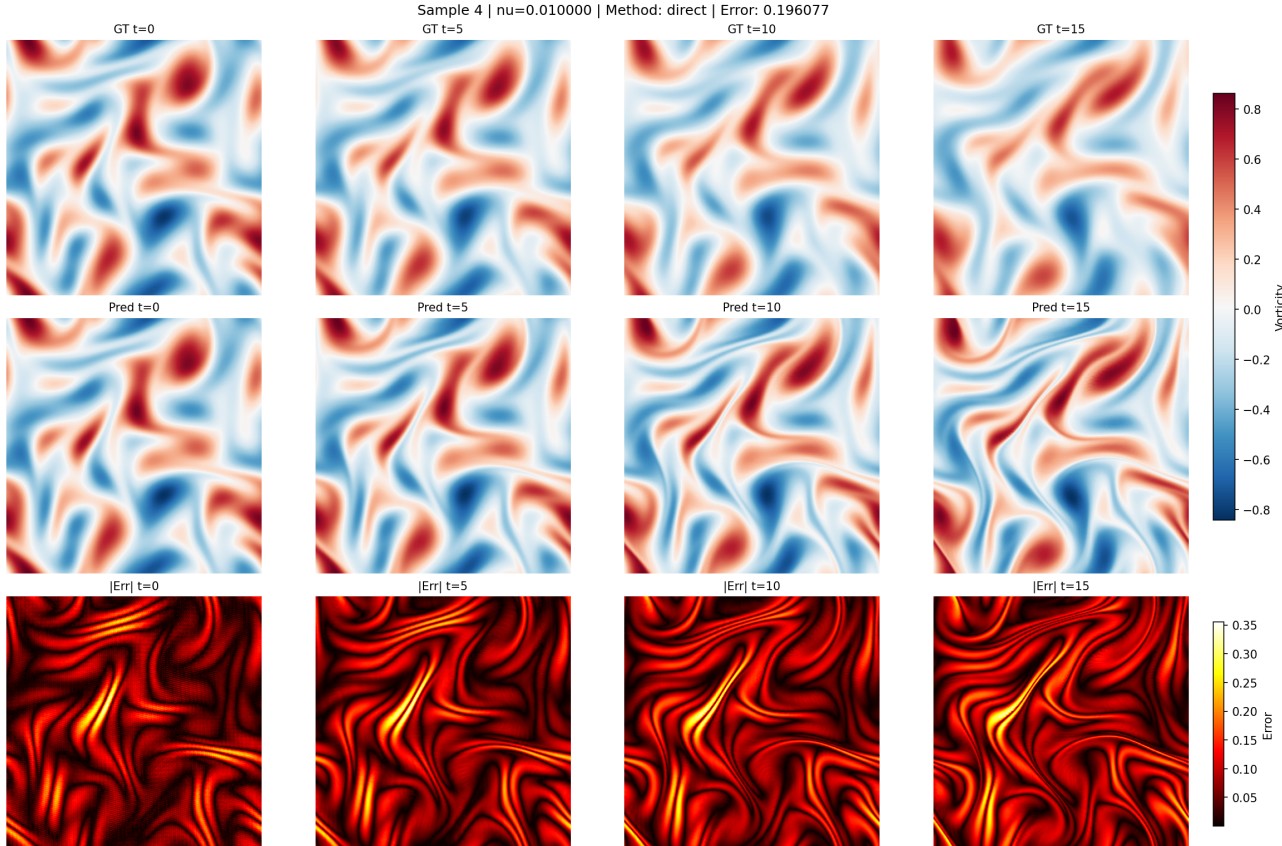

*Figure 15.* **OOD trajectory prediction on 2D Navier–Stokes.** Direct DISCO prediction with viscosity $\nu = 10^{-2}$.

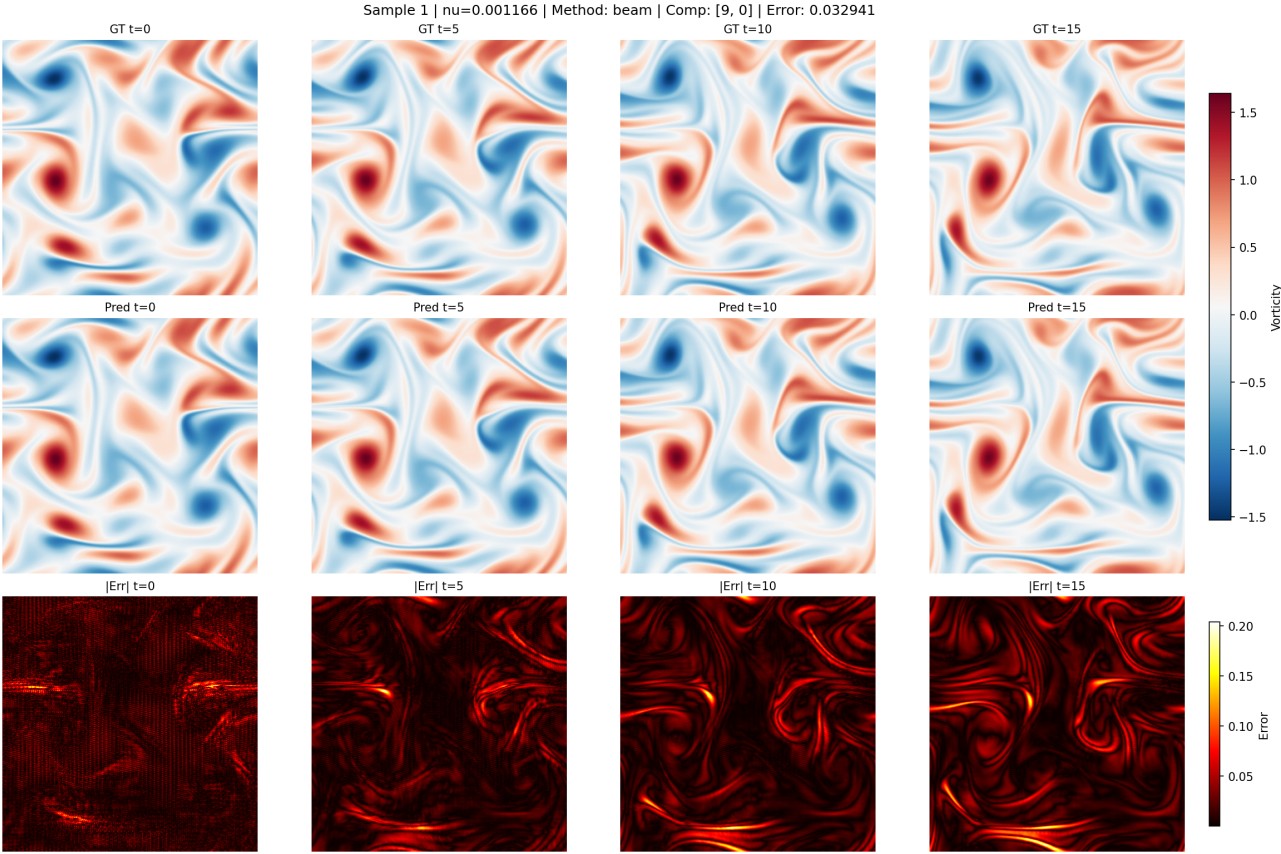

*Figure 16.* **OOD trajectory prediction on 2D Navier–Stokes.** Beam search composition with viscosity $\nu = 10^{-3}$.

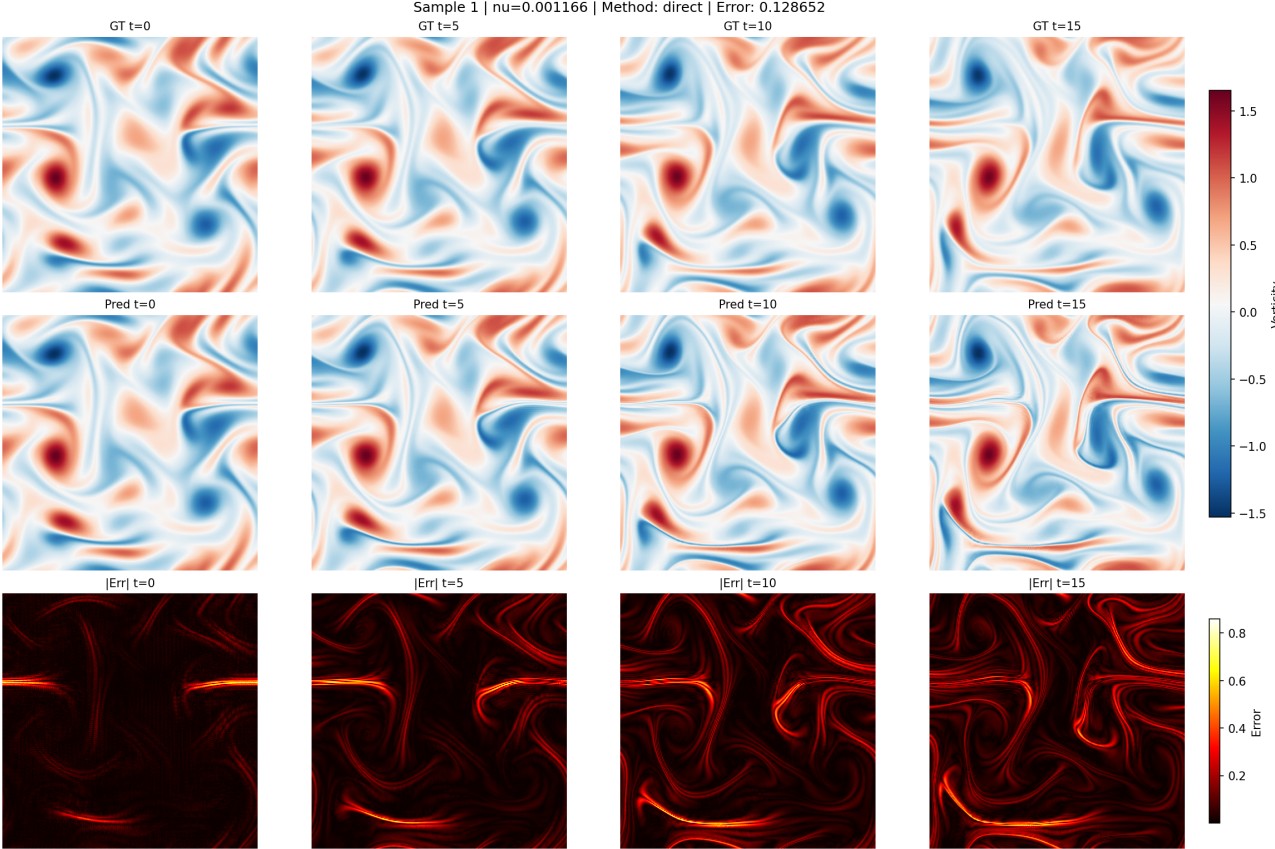

*Figure 17.* **OOD trajectory prediction on 2D Navier–Stokes.** Direct DISCO prediction with viscosity $\nu = 10^{-3}$.

