# OpenReview forum: "Test-time Generalization for Physics through Neural Operator Splitting"
_ICML.cc/2026/Conference — ICML 2026 regular_

### Official Review · Reviewer_UJ7L · 2026-03-05

**Soundness:** 3
**Presentation:** 4
**Significance:** 3
**Originality:** 3
**Overall Recommendation:** 4
**Confidence:** 3

**Summary:**

The paper introduces a method for improving the test-time generalization of neural operators for Partial Differential Equations (PDEs) without requiring any weight updates. Recognizing that standard neural operators struggle with out-of-distribution (OOD) data—such as novel physics combinations or coefficient extrapolation—the authors propose a Neural Operator Splitting strategy. They leverage the DISCO framework to learn a dictionary of basic operators during pretraining. At test time, when faced with an OOD trajectory, the method uses discrete search strategies (Uniform or Beam Search) to find an optimal subset of these pretrained operators that can approximate the novel dynamics. These operators are then sequentially combined using classical numerical analysis techniques (Lie or Strang splitting) to evolve the system. Evaluated on 1D and 2D tasks like Gray-Scott and Navier-Stokes, the method achieves state-of-the-art zero-shot generalization while enabling physical parameter estimation.

**Compliance With Llm Reviewing Policy:**

Affirmed.

**Key Questions For Authors:**

1. Computational Scaling to 3D: While the evaluations on 1D and 2D PDEs (like Gray-Scott and Navier-Stokes) convincingly demonstrate the power of operator composition, how does the computational cost of the Beam Search scale if this method were applied to 3D physical systems? Is the current sequential search strategy computationally viable when individual operator evaluations are heavily parameterized?
2. Dependency on "Pure" Training Data: The problem setting explicitly assumes parameters are drawn from a sparse distribution where only one operator is present per training trajectory. If the training data contains complex mixtures of physics where single operators cannot be easily disentangled, does the dictionary approach break down?
3. Wall-Clock Time vs. Fine-Tuning: Figure 5 presents computational efficiency in terms of FLOPs. Could you provide a direct wall-clock time comparison between your Beam Search inference and gradient-based adaptation methods like GEPS? Is the discrete search across O(BN) candidates significantly faster than taking a few LoRA gradient steps?

**Limitations:**

Yes. The authors acknowledge limitations in the Conclusion, noting the strict requirement that composed operators must share compatible input and output domains. However, they should also explicitly discuss the computational limitations and wall-clock overhead of the Beam Search mechanism, particularly regarding how this "tool-box" approach would scale to 3D environments or higher-resolution grids.

**Strengths And Weaknesses:**

Originality:

    Weakness (Tool-Box Approach): The proposed method resembles a "tool-box" assembly of existing components (DISCO for the operator dictionary, Beam Search for inference optimization, and classical Strang Splitting for temporal evolution). It lacks the elegance of a unified, end-to-end theoretical framework for OOD generalization.
    Strength (Creative Combination): Despite lacking unified mathematical elegance, the specific combination of these tools to solve combinatorial generalization in physics is highly creative. Shifting from gradient-based test-time fine-tuning to discrete search over a basis of operators is a novel perspective in Scientific Machine Learning (SciML).

Soundness:

    Strength (Rigorous OOD Evaluation): The experimental setup tests extreme OOD scenarios: Parameter Extrapolation (e.g., advection speeds 3x beyond the training hull) and Physics Composition (e.g., training on pure advection and diffusion separately, and testing on the turbulent interactions of Navier-Stokes). The inclusion of Test-Time Scaling Laws (Figure 3) provides excellent quantitative evidence that scaling search compute reduces prediction errors.
    Weakness (Dimensionality Limitations): The evaluations are strictly restricted to 1D and 2D PDEs. While 2D Navier-Stokes is complex, the computational cost of Beam Search (evaluating O(BN) operator compositions per step) may become prohibitive when moving to 3D physical systems where individual neural operator evaluations are significantly more expensive.
    Weakness (Pure Training Data Assumption): The framework relies on the assumption that training data can be isolated into trajectories of single, pure physical operators (e.g., pure advection). It is unclear if the dictionary construction remains sound if the underlying training data is fundamentally entangled.

Significance:

    Strength: The ability to generalize zero-shot to complex PDEs addresses a major bottleneck in AI for Science. The framework naturally affords System Identification, allowing practitioners to accurately estimate the unknown PDE coefficients of the target system (Figure 3, Right), offering high interpretability.
    Weakness: The current requirement that composed operators must share exactly compatible input and output domains restricts the method from mixing different physical fields or discretizations, limiting its broader "plug-and-play" utility.

Presentation:

    Strength: The paper is written with high clarity. Figure 1 provides an excellent visual summary, and the distinction between the search strategies and their complexity is articulated well.

---

> ### Author Rebuttal · Authors · 2026-03-31
>
> We thank the reviewer for the positive assessment and the thoughtful questions.
>
> **Q1.** The search cost is O(B×N×L) neural operator forward passes per iteration, regardless of spatial dimension. What changes in 3D is the cost of each individual forward pass, but the ratio of search overhead to total inference cost remains the same. Candidates are also independent and can be evaluated in parallel, so the search overhead does not fundamentally limit applicability in higher dimensions. It will be an interesting direction for future work to further study the proposed strategy for 3D systems.
>
> **Q2.** We retrained DISCO where training trajectories contain both advection and diffusion simultaneously (Mixed 0.5: 50% coupled, 25% pure advection, 25% pure diffusion; Mixed 1.0: 100% coupled) and evaluate on joint parameter extrapolation with both coefficients in [0.01, 2]. With Mixed 0.5 the dictionary still contains some pure operators; Mixed 1.0 contains only coupled ones. Beam search gains scale with dictionary size, at small dictionary size there are not enough diverse compositions for the search to be effective, but with larger dictionaries the method recovers meaningful improvements even from fully mixed operators:
>
> | Model | # operators | Direct | Beam | Improvement |
> | --- | --- | --- | --- | --- |
> | Mixed 0.5 | 64 | 0.098 | 0.097 | 0% |
> | Mixed 0.5 | 256 | 0.097 | 0.057 | 41% |
> | Mixed 1.0 | 64 | 0.313 | 0.267 | 15% |
> | Mixed 1.0 | 256 | 0.278 | 0.190 | 32% |
>
> **Q3.** We measured per-sample wall-clock time for GEPS finetuning and our approach on a single A100:
>
> | Method | Finetuning Steps | advection + diffusion NRMSE / Time | Extrapolation advection NRMSE / Time | Extrapolation diffusion NRMSE / Time |
> | --- | --- | --- | --- | --- |
> | GEPS | 100 | 0.265 / ~3.5s | 1.129 / ~3.5s | 0.599 / ~3.5s |
> | GEPS | 500 | 0.103 / ~15s | 0.847 / ~15s | 0.268 / ~15s |
> | GEPS | 2000 | 0.034 / ~57s | Inf (diverged) | Inf (diverged) |
> | Ours (Beam) | — | **0.015 / ~21s** | **0.052 / ~26s** | **0.002 / ~26s** |
>
> Our method is faster than GEPS-2000 on advection-diffusion and twice more accurate. On extrapolation, GEPS at 2000 steps diverges entirely while our method remains stable and performant. Increasing GEPS compute does not help in far-OOD regimes, while our approach reliably benefits from more search (Figure 5).

---

> > ### Author Rebuttal · Reviewer_UJ7L · 2026-04-01
> >
> > I thank the authors for the detailed rebuttal. The wall-clock timing comparisons with GEPS, the dictionary subsampling sensitivity analysis, and the additional perturbation experiments (ε variations) satisfactorily address my concerns. The theoretical justification for coefficient estimation via Lie/Strang splitting error analysis is appreciated. The commitment to add figure clarifications and FNO baseline strengthens the paper.

---

### Official Review · Reviewer_SLt5 · 2026-03-10

**Soundness:** 3
**Presentation:** 4
**Significance:** 3
**Originality:** 3
**Overall Recommendation:** 5
**Confidence:** 4

**Summary:**

This paper studies test-time generalization for PDE forecasting under out-of-distribution physics. Building on DISCO, it constructs a dictionary of neural operators from training trajectories, and at test time searches for operator compositions that best match the observed trajectory. Experiments on several PDE benchmarks show improved performance over in-context learning, pretraining-based, and meta-learning baselines.

**Compliance With Llm Reviewing Policy:**

Affirmed.

**Final Justification:**

My primary concerns have now been satisfactorily addressed, and the additional experimental results make the paper stronger.

**Key Questions For Authors:**

1. The experimental setting is relatively idealized: training is restricted to single-operator trajectories, while composed dynamics only appear at test time. In more realistic settings, training data often already contain coupled dynamics, and different operators may not be strictly isolated. Can the method generalize to a more realistic multi-physics foundation model setting?

2. What is the concrete computational cost of the test-time search procedure? Since the method introduces additional inference-time computation, it would be useful to clarify whether this limits its applicability in scenarios with strict efficiency or latency requirements.

3. How long does the fitting window at test time need to be in practice? It would strengthen the paper to include an analysis of how performance changes when the fitting window is too short or too long.

**Limitations:**

yes

**Strengths And Weaknesses:**

## Strengths
1. The paper addresses test-time generalization, which is an important and practically relevant problem. In real-world settings, training data are unlikely to fully cover all possible physical regimes, so improving generalization at test time is valuable.

2. The method is technically sound and well motivated. Building on a DISCO-pretrained operator dictionary, the paper performs test-time beam search over operator compositions and uses neural operator splitting to approximate multi-physics dynamics in a principled way.

3. The experimental results are solid. Across the reported benchmarks, the method shows clear improvements over in-context learning, pretraining-based, and meta-learning baselines.

---


## Weaknesses
1. The method relies heavily on DISCO. It is therefore unclear how well the proposed test-time search and composition strategy would generalize to other in-context learning or neural operator frameworks.

2. The experimental setting is somewhat idealized. In the experiments, training trajectories are deliberately restricted to single-operator dynamics, while test-time trajectories contain composed operators. In more realistic settings, training data often already include coupled dynamics, and different operators may not be so cleanly isolated. It remains unclear whether the method would generalize to a more realistic multi-physics foundation model setting. Relatedly, it would be useful to understand whether the method becomes unstable when the train-test gap is larger, and whether there are representative failure cases.

3. The computational cost of test-time search is not fully clear. Since the method performs search at inference time, it may be less suitable for applications with strict computation or latency constraints.

4. The paper does not sufficiently analyze the fitting window length used at test time. It would be helpful to understand how long a fitting window is needed in practice, and how performance changes when the window is too short or too long.

---

> ### Author Rebuttal · Authors · 2026-03-31
>
> We thank the reviewer for the constructive feedback on our manuscript.
>
> **W1.** The search and splitting strategy are architecture-agnostic. What is required is a framework that produces neural operators conditioned on trajectory context and formulates them as neural ODEs so that classical splitting applies. DISCO satisfies both and is state-of-the-art, but other neural ODE-based architectures would be compatible. We will clarify this in the paper.
>
> **W2 / Q1.** We ran two stress tests.
>
> *Test 1 — unseen term.* Recall that training data consists purely of advection or diffusion trajectories. We introduce a nonlinear Burgers-like term at test time: ∂u/∂t + v ∂u/∂x + ε · u · ∂u/∂x = D ∂²u/∂x², where ε=0 recovers standard OOD advection-diffusion and ε>0 introduces a term not representable by any dictionary operator. The method degrades gracefully: performance is essentially unchanged at small ε, deteriorates smoothly at larger values, and beam search strictly outperforms direct prediction throughout. The advection coefficient is correctly identified across all ε, even when the dynamics are no longer decomposable. We compute the metrics over 84 rollout steps, with advection speed v=0.5, and diffusion D=0.3.
>
> | ε | Direct NRMSE | Beam NRMSE | Est. v | Est. D |
> | --- | --- | --- | --- | --- |
> | 0.00 | 0.319 | **0.055** | 0.495 | 0.316 |
> | 0.01 | 0.323 | **0.064** | 0.507 | 0.328 |
> | 0.05 | 0.355 | **0.061** | 0.518 | 0.315 |
> | 0.10 | 0.417 | **0.089** | 0.511 | 0.279 |
> | 0.25 | 0.704 | **0.184** | 0.502 | 0.276 |
> | 0.50 | 0.843 | **0.414** | 0.496 | 0.125 |
> | 1.00 | 0.817 | **0.797** | 0.496 | 0.048 |
>
> *Test 2 — mixed training.* We retrained DISCO where training trajectories contain both advection and diffusion simultaneously (Mixed 0.5: 50% coupled, 25% pure advection, 25% pure diffusion; Mixed 1.0: 100% coupled) and evaluate on joint parameter extrapolation with both coefficients in [0.01, 2]. With Mixed 0.5 the dictionary still contains some pure operators; Mixed 1.0 contains only coupled ones. Beam search gains scale with dictionary size, at small dictionary size there are not enough diverse compositions for the search to be effective, but with larger dictionaries the method recovers meaningful improvements even from fully mixed operators:
>
> | Model | # operators | Direct | Beam | Improvement |
> | --- | --- | --- | --- | --- |
> | Mixed 0.5 | 64 | 0.098 | 0.097 | 0% |
> | Mixed 0.5 | 256 | 0.097 | 0.057 | 41% |
> | Mixed 1.0 | 64 | 0.313 | 0.267 | 15% |
> | Mixed 1.0 | 256 | 0.278 | 0.190 | 32% |
>
> **W3 / Q2.** **Q3.** We measured per-sample wall-clock time for GEPS finetuning and our approach on a single A100:
>
> | Method | Finetuning Steps | advection + diffusion NRMSE / Time | Extrapolation advection NRMSE / Time | Extrapolation diffusion NRMSE / Time |
> | --- | --- | --- | --- | --- |
> | GEPS | 100 | 0.265 / ~3.5s | 1.129 / ~3.5s | 0.599 / ~3.5s |
> | GEPS | 500 | 0.103 / ~15s | 0.847 / ~15s | 0.268 / ~15s |
> | GEPS | 2000 | 0.034 / ~57s | NaN (diverged) | Inf (diverged) |
> | Ours (Beam) | — | **0.015 / ~21s** | **0.052 / ~26s** | **0.002 / ~26s** |
>
> Our method is faster than GEPS-2000 on advection-diffusion and twice more accurate. On extrapolation, GEPS at 2000 steps diverges entirely while our method remains stable and performant. Increasing GEPS compute does not help in far-OOD regimes, while our approach reliably benefits from more search (Figure 5).
>
> **W4 / Q3.** We swept L - the number of available snapshots for objective optimization - in {2, 4, 8} on advection+diffusion:
>
> | L | Beam NRMSE |
> | --- | --- |
> | 2 | **0.011** |
> | 4 | 0.013 |
> | 8 | 0.013 |
> | 16 (paper default) | ~0.015 |
>
> The method is robust on advection diffusion, but we expect this to depend on the dataset. This suggests that we could potentially reduce the window used to compute the objective to speed up the search.

---

> > ### Author Rebuttal · Reviewer_SLt5 · 2026-04-02
> >
> > My main concerns have been addressed in the rebuttal. The additional results substantially strengthen the paper, and the new empirical evidence makes the overall case much more convincing. Based on these clarifications and results, I am inclined to raise my score to 5. I would also encourage the authors to incorporate these results into the paper.

---

### Official Review · Reviewer_RRZc · 2026-03-10

**Soundness:** 3
**Presentation:** 3
**Significance:** 4
**Originality:** 4
**Overall Recommendation:** 5
**Confidence:** 4

**Summary:**

The authors present a test-time compute framework for neural operators. Building on DISCO, it performs beam search over a set of neural operators, each pretrained on a particular fundamental physics operator, and composes them via operator splitting to handle unseen PDE coefficients and unseen combinations of physical effects. The method can also identify the underlying coefficients from the selected operators.

**Compliance With Llm Reviewing Policy:**

Affirmed.

**Final Justification:**

This is a strong paper, and the author's rebuttals strengthened the confidence i have in their empirical results and presentation.

**Key Questions For Authors:**

1. The paper does not address whether the proposed splitting procedure preserves the resolution-invariance properties often associated with neural operators. Is zero-shot super-resolution retained, or is the method only supported for matched discretizations?
2. Test-time fitting minimizes $\frac{1}{L-1} \sum_{t=1}^{L-1} \operatorname{NRMSE}(u_{t+1}^\text{test}, \hat{u}_{t+1}^\text{test})$, whereas the main reported metric is rollout NRMSE over the predicted trajectory. Why was a one-step fitting objective chosen, rather than a multi-step or trajectory-level objective that is better aligned with evaluation?
3. The paper estimates PDE coefficients by summing the contributions of the selected dictionary elements. Given that prediction is performed by composing learned operators via Lie/Strang splitting, is this additive coefficient estimate theoretically justified?
4. Is there some order of error that can be derived for the summed parameter estimate?
5. I understand that some columns in Table 1 can be related to certain PDEs (such as Navier-Stokes), however, can the remaining columns be related to others (e.g., Burgers, Kuramoto–Sivashinsky, etc)?

If questions 2 and 3/4 are resolved, I would be open to increasing my score.

**Limitations:**

Wall clock and FLOP cost of the test-time fitting is underdiscussed. Otherwise limitations are adequately considered.

**Strengths And Weaknesses:**

### Strengths
* The approach presented is novel and addresses a very pertinent issue in neural operators: cross-equation generalization.
* The mathematical foundations of the model (operator splitting) is very well explicated
* The method of parameter estimation is clever, and very positive for the interpretability of the proposed method

### Weaknesses
* The computational comparison is incomplete. Although the paper reports fitting error versus cumulative FLOPs for beam and uniform search, it does not provide matched training- or test-time wallclock/FLOPs across baselines. Since the proposed method introduces an explicit test-time search procedure, it remains unclear whether the accuracy gains come at substantially greater inference cost.
* Section 4 presents a dictionary built from training trajectories $\{ f_{1},\dots f_{N} \}$, but Section 5 later says beam search uses subsampled N values such as 256, 96, 40, and 17 depending on the equation. It is not clear what the full dictionary size was, how subsampling was done, or how sensitive results are to that choice.
* The prediction error metric in figure 3 (line 289)  and table 3 metric (line 825) are not reported. Could the authors please clarify what these are, possibly in the table legends.
* The baseline suite excludes many classic NO models, which are still commonplace, especially in subfields such as SPDE. FNO, Galerkin transformer, etc.
* The legend in Figure 2 appears contradictory to the figure content. The legend states the fourth row is pure diffusion, yet it is labelled "Diffusion + kill dynamics" in the figure itself.
* Figure 2 is somewhat unclear. For the Gray-Scott experiment, the paper states that evaluation uses $L=16$ context snapshots followed by rollout prediction, but the figure does not indicate which displayed times are conditioning frames and which are actual test-time predictions. Since the reaction-diffusion trajectory is obtained by combining pretrained operators at test time, it would help to mark the context/prediction split explicitly.

The authors have produced some excellent and insightful work here to which I see immediate application to many other subdomains of operator learning. I hope the authors make their code public upon release.

---

> ### Author Rebuttal · Authors · 2026-03-31
>
> We thank the reviewer for the detailed and constructive feedback and for the positive assessment.
>
> **W1.** We measured per-sample wall-clock time for GEPS finetuning and our approach on a single A100:
>
> | Method | Finetuning Steps | advection + diffusion NRMSE / Time | Extrapolation advection NRMSE / Time | Extrapolation diffusion NRMSE / Time |
> | --- | --- | --- | --- | --- |
> | GEPS | 100 | 0.265 / ~3.5s | 1.129 / ~3.5s | 0.599 / ~3.5s |
> | GEPS | 500 | 0.103 / ~15s | 0.847 / ~15s | 0.268 / ~15s |
> | GEPS | 2000 | 0.034 / ~57s | Inf (diverged) | Inf (diverged) |
> | Ours (Beam) | — | **0.015 / ~21s** | **0.052 / ~26s** | **0.002 / ~26s** |
>
> Our method is faster than GEPS-2000 on advection-diffusion and twice more accurate. On extrapolation, GEPS at 2000 steps diverges entirely while our method remains stable and performant. Increasing GEPS compute does not help in far-OOD regimes, while our approach reliably benefits from more search (Figure 5).
>
> **W2.** In most experiments, training trajectories are grouped by environment (parameter configuration). We select one representative operator per environment, ensuring the dictionary spans the full range of training dynamics without redundancy. When a model is trained with a codebook, we take the corresponding environment code. A sweep over the number N of operators in the dictionary in {16, 32, 64, 256} on advection+diffusion shows:
>
> | # operators | Beam NRMSE |
> | --- | --- |
> | 16 | 0.037 |
> | 32 | 0.035 |
> | 64 | 0.017 |
> | 256 (paper) | **0.015** |
>
> Performance is stable at small N and stabilizes around 64 operators, confirming robustness to the subsampling choice.
>
> **W3.** The prediction error in Figure 3 is rollout NRMSE averaged over 34 steps. The metric in Table 3 is MAE between estimated and true PDE coefficients. We will add these to the captions.
>
> **W4.** We will include an FNO baseline in the camera-ready version. For this rebuttal we focused on methods with test-time adaptation, as a vanilla FNO without adaptation is expected to behave similarly to MPP.
>
> **W5 / W6.** We will fix the Figure 2 legend and add a clear marker at t=16 separating the L=16 context frames from the predicted rollout.
>
> **Q1.** Splitting is purely temporal and resolution-agnostic by construction. Whether the composed system is resolution-invariant depends entirely on the backbone (DISCO's U-Net), not on the splitting procedure. If the backbone is resolution-invariant, so is the composed system.
>
> **Q2/Q3.** The one-step objective averages NRMSE over L-1 consecutive one-step transitions, all of which can be evaluated in parallel across the context window. A multi-step rollout with K steps instead requires K sequential forward passes per candidate, making it K times more time consuming at the same dictionary size. Since we evaluate O(B×N) candidates per beam search iteration, keeping per-candidate cost low directly multiplies the search coverage we can afford at a fixed compute budget. Figure 3 (left) shows that one-step fitting error correlates strongly with rollout NRMSE, justifying its use as a proxy for the evaluation metric. However, using a multiple step objective might be useful to select operators that are the most stable over time and is an interesting direction.
>
> **Q4/Q5.** For linear operators, the Lie splitting flow satisfies $e^{Δt(F₁+F₂)} = e^{ΔtF₁} \circ e^{ΔtF₂} + O(Δt²)$, so if operator i approximates μᵢ·Fᵢ, the composed system approximates $Σμᵢ·Fi$ as $Δt→0$. The estimate is therefore exact for linear operators and a first-order approximation (O(Δt) for Lie, O(Δt²) for Strang) in the general case. Furthermore, the perturbation experiment (see our response to W1 from Reviewer **rJBq**) provides supporting evidence that coefficients can be recovered partially (advection speed in this case) even when the dynamics are no longer exactly decomposable into operators seen during training.
>
> **Q6.** Yes, the combined equation columns correspond to: Nonlinear Adv.+Diff. → Burgers-like, Nonlinear Adv.+Dispersion → KdV-like,  Reaction+Diff. → Gray-Scott, Euler+Diff. → Navier-Stokes. We will add these labels to Table 1.

---

> > ### Author Rebuttal · Reviewer_RRZc · 2026-04-01
> >
> > I thank the authors for their prompt response. The benchmark experiments and clarifications satisfactorily address my concerns. My only remaining suggestion is to read over the figure and table legends once more to ensure the experiments are fully clear, since this was the primary clarity issue in the original submission.

---

### Official Review · Reviewer_rJBq · 2026-03-17

**Soundness:** 2
**Presentation:** 3
**Significance:** 3
**Originality:** 3
**Overall Recommendation:** 4
**Confidence:** 3

**Summary:**

This paper proposes a test-time adaptation strategy for neural PDE surrogates that composes pretrained operators via classical operator splitting (Lie/Strang) to handle out-of-distribution dynamics. Built on DISCO (which learns a dictionary of neural operators from training trajectories), the method searches at test time—via beam search or uniform sampling—over subsets of these dictionary operators whose splitting-based composition best fits an observed OOD trajectory. The approach is evaluated on four benchmarks (1D advection-diffusion, 1D combined equation, 2D Gray–Scott, 2D Navier–Stokes) under parameter extrapolation and physics composition scenarios, showing substantial improvements over baselines.

**Compliance With Llm Reviewing Policy:**

Affirmed.

**Final Justification:**

I maintain my recommendation of weak accept.

**Key Questions For Authors:**

See the weaknesses.

**Limitations:**

yes

**Strengths And Weaknesses:**

Strengths:

1. The separation of training data into isolated single-physics trajectories and the two OOD test settings (parameter extrapolation vs. operator composition) is clean. This makes the experimental protocol easy to evaluate and the claims testable.

2. Borrowing Lie and Strang splitting from classical PDE numerics and applying it to learned neural operators is a natural and appealing idea. The paper correctly notes the error order distinction between Lie (O(Δt²)) and Strang (O(Δt³)) splitting, and the extension to m operators in Appendix B.3 is properly handled.

Weaknesses:

1. The assumption is very restrictive. The entire method hinges on the assumption that OOD dynamics can be expressed as $\partial_t u = \sum \mu_k F_k$, where $F_k$ are the *same* operators seen during training, just combined. This is a strong structural prior. In practice, many interesting OOD scenarios involve genuinely new physics (e.g., turbulence closure terms, phase transitions, chemical kinetics with different functional forms) that cannot be decomposed into a sum of previously seen terms. The paper acknowledges the limitation of compatible input/output domains in the conclusion, but it is still cannot be ignored.

2. The experimental setup is heavily tailored to the method's assumptions. Every benchmark is constructed so that the test-time dynamics are *exactly* sums of training dynamics. The advection-diffusion test is literally advection + diffusion; the Navier–Stokes test is Euler + heat equation. This is by design, but it means the experiments don't stress-test the method's limits. What happens when the composition isn't clean? For instance, in Navier–Stokes, the nonlinear advection term couples to viscosity through the velocity field—the dynamics aren't truly separable into independent Euler and diffusion components. The fact that the method still works reasonably suggests robustness, but this isn't analyzed or discussed. A controlled experiment with "imperfect decomposability" (e.g., adding a small cross-term that cannot be represented by any dictionary operator) would significantly strengthen the paper.

3. Operator splitting error analysis is incomplete for the neural case.  Classical splitting error bounds assume exact sub-operators. Here, each $f_i$ is an *approximate* learned operator with its own error $\varepsilon_i$ . The paper shows empirically in Table 3 (Appendix C.1) that the composed error tracks the worst individual operator, but there's no formal analysis of how splitting error and learning error interact. In particular, are there pathological cases where small individual errors compound through splitting? The single experiment in Table 3 (heat + dispersion only) is insufficient to make general claims about this interaction.

4. Baseline comparisons have caveats. All baselines are "trained from scratch on the same training datasets designed for this study." This is fair in one sense, but the training setup (strict single-physics isolation) is atypical. MPP and Zebra were designed for multi-physics pretraining across diverse datasets; training them on isolated single-physics data likely underperforms their intended use case. The comparison would be more convincing if the authors also showed results where baselines are trained on their *intended* data distributions (including mixed physics) and then evaluated on the same OOD tasks.


Minor Issues:

1. How does the method perform when the test dynamics contain a component *not* representable by any dictionary operator (e.g., a forcing term or a new nonlinearity)?
2. Can you provide an ablation isolating the effect of the modified training recipe (Section B.2) from the test-time search?
3. How sensitive are results to the dictionary subsampling strategy used before beam search?

---

> ### Author Rebuttal · Authors · 2026-03-31
>
> We thank the reviewer for the detailed feedback. We address each point below.
>
> **W1/W2/Q1** We have added an additional experiment to show how the beam search behaves when the test dynamics also includes a derivative term (nonlinear advection) that was never seen during training. Specifically, we ran the following additional experiment. Recall that in the advection-diffusion setting, the training data consists purely of advection or diffusion dynamics. We consider a test dynamics with an additional unseen  nonlinear term (Burgers-like): ∂u/∂t + v ∂u/∂x + ε · u · ∂u/∂x = D ∂²u/∂x², where ε=0 recovers the advection + diffusion OOD test case used in the main paper experiments. Meanwhile, ε>0 corresponds to a more challenging setting that lies beyond the regime of exact composition of the training dynamics. Results are obtained for v=0.5, D=0.3, with 128 test trajectories per ε, and we rollout the system for 84 steps:
>
> | ε | Direct NRMSE | Beam NRMSE | Est. v | Est. D |
> | --- | --- | --- | --- | --- |
> | 0.00 | 0.319 | **0.055** | 0.495 | 0.316 |
> | 0.01 | 0.323 | **0.064** | 0.507 | 0.328 |
> | 0.05 | 0.355 | **0.061** | 0.518 | 0.315 |
> | 0.10 | 0.417 | **0.089** | 0.511 | 0.279 |
> | 0.25 | 0.704 | **0.184** | 0.502 | 0.276 |
> | 0.50 | 0.843 | **0.414** | 0.496 | 0.125 |
> | 1.00 | 0.817 | **0.797** | 0.496 | 0.048 |
>
> The degradation is graceful: at ε ≤ 0.05 performance is essentially unaffected, and beam search strictly outperforms direct prediction at every ε. Notably, the advection coefficient v is recovered correctly across all ε values, while the diffusion estimate degrades at large ε.
>
> **W3.** Table 3 shows that the composed error is dominated by the least accurate individual operator on diffusion + dispersion, without observing compounding errors. We expect settings where composition will not work regardless of any formal guarantee, for instance, when operators are trained on different initial condition distributions and their intermediate states become incompatible.
>
> **W4.** These methods (MPP, Zebra) can be pretrained on multiple physics or trained with a parametric setting, which is the setting we adopt here. Training MPP or Zebra on mixed-physics data would expose them to the test distribution, and would not be compatible with our evaluation protocol.
>
> **Q2.** We ran the ablation by training without in-context learning (advection / diffusion) and without the codebook (Navier-Stokes).
>
> | Setting | Paper (with context or codebook) | No-context / No-codebook |
> | --- | --- | --- |
> |  | Beam | Beam |
> | advection + diffusion | 0.015 | 0.026 |
> | Advection extrap. | 0.052 | 0.406 |
> | Diffusion extrap. | 0.002 | 0.012 |
> | Navier-Stokes | 0.066 | 0.563 |
>
> Both components improve operator search and compositionality. The modified training recipe is especially important for far-OOD extrapolation, while test-time search also provides a consistent gain. Without the improved training recipe, the operators tend to be specific to the initial conditions of their training trajectories and transfer poorly.
>
> **Q3.** In most experiments, training trajectories are grouped by environment (parameter configuration). We select one representative operator per environment, ensuring the dictionary spans the full range of training dynamics without redundancy. When a model is trained with a codebook, we take the corresponding environment code. A sweep over the number N of operators in the dictionary in {16, 32, 64, 256} on advection+diffusion shows:
>
> | # operators | Beam NRMSE |
> | --- | --- |
> | 16 | 0.037 |
> | 32 | 0.035 |
> | 64 | 0.017 |
> | 256 (paper) | **0.015** |
>
> Performance is stable at small N and stabilizes around 64 operators, confirming robustness to the subsampling choice.

---

> > ### Author Rebuttal · Reviewer_rJBq · 2026-04-04
> >
> > I thank the authors for their rebuttal. I keep my positive rating.

---

### Decision · Program_Chairs · 2026-04-30

**Decision:**

Accept (regular)

**Comment:**

This paper proposes a test-time generalization framework for neural operators that composes pretrained operators via classical operator splitting to handle out-of-distribution dynamics without parameter updates. The idea is original and technically well grounded, and the paper addresses an important problem in scientific machine learning: zero-shot generalization across unseen PDE coefficients and novel combinations of physics. Reviewers found the method novel, well motivated, and empirically strong, with particular strengths in cross-equation generalization and interpretability through coefficient recovery.

The main initial concerns were the restrictive compositional assumption, the idealized nature of the experimental setup, and the computational overhead of test-time search. In the rebuttal, the authors addressed these points well by providing additional experiments on unseen perturbation terms, mixed-physics training settings, dictionary subsampling, fitting-window sensitivity, and wall-clock comparisons against adaptation baselines. These new results substantially strengthen the empirical case and clarify the scope of the claims.

Overall, the reviewer consensus after rebuttal is positive, and the concerns were largely resolved.